# Value-based attentional capture affects multi-alternative decision making

Sebastian Gluth[1†]*, Mikhail S Spektor[1,2†], Jörg Rieskamp[1]

[1]Department of Psychology, University of Basel, Basel, Switzerland; [2]Department of Psychology, University of Freiburg, Freiburg, Germany

**Abstract** Humans and other animals often violate economic principles when choosing between multiple alternatives, but the underlying neurocognitive mechanisms remain elusive. A robust finding is that adding a third option can alter the relative preference for the original alternatives, but studies disagree on whether the third option's value decreases or increases accuracy. To shed light on this controversy, we used and extended the paradigm of one study reporting a positive effect. However, our four experiments with 147 human participants and a reanalysis of the original data revealed that the positive effect is neither replicable nor reproducible. In contrast, our behavioral and eye-tracking results are best explained by assuming that the third option's value captures attention and thereby impedes accuracy. We propose a computational model that accounts for the complex interplay of value, attention, and choice. Our theory explains how choice sets and environments influence the neurocognitive processes of multi-alternative decision making.
DOI: https://doi.org/10.7554/eLife.39659.001

## Introduction

In recent years, studying choices between multiple (i.e., more than two) alternatives has attracted growing interest in many research areas including biology, neuroscience, psychology, and economics (*Berkowitsch et al., 2014*; *Chau et al., 2014*; *Chung et al., 2017*; *Cohen and Santos, 2017*; *Gluth et al., 2017*; *Hunt et al., 2014*; *Landry and Webb, 2017*; *Lea and Ryan, 2015*; *Louie et al., 2013*; *Mohr et al., 2017*; *Spektor et al., 2018a*; *Spektor et al., 2018b*). In such choice settings humans and other animals often violate the independence from irrelevant alternatives (IIA) principle of classical economic decision theory (*Rieskamp et al., 2006*). This principle states that the relative preference for two options must not depend on a third (or any other) option in the choice set (*Luce, 1959*; *Marschak and Roy, 1954*). Violations of IIA have profound implications for our understanding of the neural and cognitive principles of decision making. For example, if choice options are not independent from each other, we cannot assume that the brain first calculates each option's value separately before selecting the option with the highest (neural) value (*Vlaev et al., 2011*). Instead, dynamic models in which option comparison and decision formation processes are intertwined appear more promising as they account for several violations of IIA (*Gluth et al., 2017*; *Hunt and Hayden, 2017*; *Hunt et al., 2014*; *Mohr et al., 2017*; *Roe et al., 2001*; *Trueblood et al., 2014*; *Tsetsos et al., 2012*; *Usher and McClelland, 2004*).

The underling mechanisms of multi-alternative choice and the conditions under which decision makers exhibit different forms of violations of IIA are a matter of current debate (*Chau et al., 2014*; *Frederick et al., 2014*; *Louie et al., 2013*; *Spektor et al., 2018a*; *Spektor et al., 2018b*). Two recent studies by Louie and colleagues (*Louie et al., 2013*; henceforth Louie2013) and by Chau and colleagues (*Chau et al., 2014*; henceforth Chau2014) investigated whether and how the value of a third option influences the *relative choice accuracy* between two other options (i.e., the probability of selecting the option with the higher value). Both studies reported violations of IIA which, however, contradicted each other: Louie2013 found a negative relationship, so that better third options

*For correspondence:
sebastian.gluth@unibas.ch

†These authors contributed equally to this work

Competing interests: The authors declare that no competing interests exist.

**eLife digest** A man in a restaurant is offered a choice between apple or blueberry pie, and chooses apple. The waiter then returns a few moments later and tells him they also have cherry pie available. "In that case", replies the man, "I'll have blueberry".

This well-known anecdote illustrates a principle in economics and psychology called the independence principle. This states that preferences between two options should not change when a third option becomes available. A person who prefers apple over blueberry pie should continue to do so regardless of whether cherry pie is also on the menu. But, as in the anecdote, people often violate the independence principle when making decisions. One example is voting. People may vote for a candidate who would not usually be their first choice only because there is also a similar but clearly less preferable candidate available.

Such behavior provides clues to the mechanisms behind making decisions. Studies show, for example, that when people have to choose between two options, introducing a desirable third option that cannot be selected – a distractor – alters what decision they make. But the studies disagree on whether the distractor improves or impairs performance.

Gluth et al. now resolve this controversy using tasks in which people had to choose between rectangles on a computer screen for the chance to win different amounts of money. Contrary to a previous study, their four experiments showed that a high-value distractor did not change how likely the volunteers were to select one of the two available options over the other. Instead, the distractor slowed down the entire decision-making process. Moreover, volunteers often selected the high-value distractor despite knowing that they could not have it. One explanation for such behavior is that high-value items capture our attention automatically even when they are irrelevant to our goals. If a person likes chocolate cake, their attention will immediately be drawn to a cake in a shop window, even if they had no plans to buy a cake. Eye-tracking data confirmed that volunteers in the above experiments spent more time looking at high-value items than low-value ones. Those volunteers whose gaze was distracted the most by high-value items also made the worst decisions.

Based on the new data, Gluth et al. developed and tested a mathematical model. The model describes how we make decisions, and how attention influences this process. It provides insights into the interplay between attention, valuation and choice – particularly when we make decisions under time pressure. Such insights may enable us to improve decision-making environments where people must choose quickly between many options. These include emergency medicine, road traffic situations, and the stock market. To achieve this goal, findings from the current study need to be tested under more naturalistic conditions.

DOI: https://doi.org/10.7554/eLife.39659.002

*decreased* the relative choice accuracy (i.e., the probability of choosing the better of the two original options). They attributed this to *divisive normalization* of (integrated) value representations in the brain: To keep value coding by neural firing rates in a feasible range, each value code could be divided by the sum of all values. Thus, a third option of higher value implies a higher division of firing rates and reduces the neural discriminability between the other two options. In contrast, Chau2014 reported a positive relationship, that is, third options of higher values *increased* relative choice accuracy. They predicted this effect by a *biophysical cortical attractor* model (*Wang, 2002*). Briefly, this model assumes pooled inhibition between competing attractors that represent accumulated evidence for the different options. A third option of higher value would increase the level of inhibition in the model and thus lead to more accurate decisions (because inhibition reduces the influence of random noise).

The goal of the present study was to resolve these opposing results by exploiting the multi-attribute nature of Chau2014's task and extending it with respect to the paradigm and the analysis of the behavioral data. Notably, the paradigms of Louie2013 and Chau2014 differ in several aspects, the most critical being the type of choice options: In Louie2013, participants decided between different food snacks whereas in Chau2014, participants chose between rectangles that represented gambles with reward magnitude and probability conveyed by the rectangles' color and orientation, respectively (*Figure 1A and B*). Our initial hypothesis for the discrepancy between the studies'

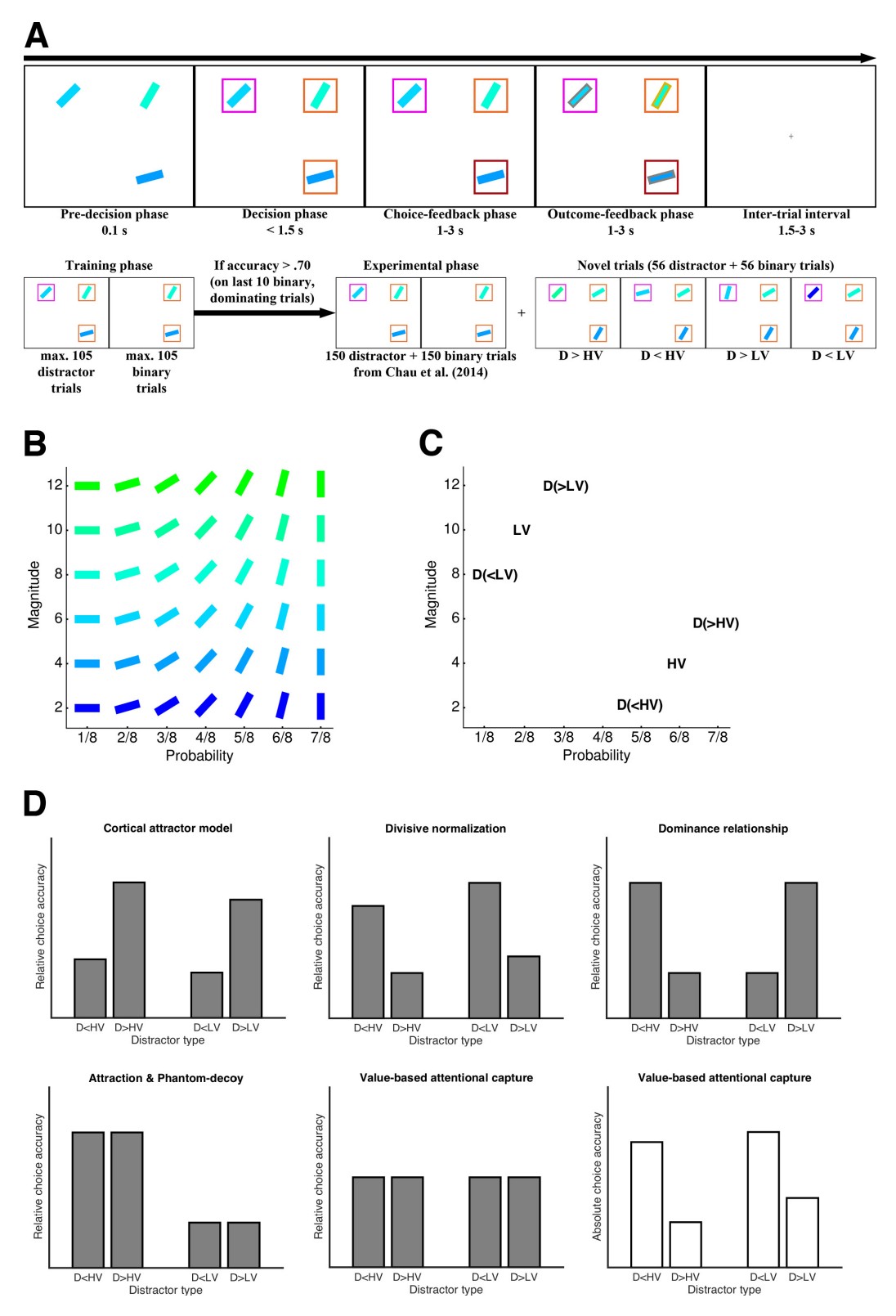

**Figure 1.** Study design and predictions for novel trials. (**A**) Example trial (upper panel) and general workflow (lower panel) of the Chau2014 paradigm as used in our four experiments (variations in specific experiments are mentioned in the main text). In the critical trials, participant had 1.6 s to choose between two rectangles with a third rectangle displayed but declared unavailable after 0.1 s. (**B**) Stimulus matrix showing one (of four possible) associations of color and orientation of rectangles with reward probability and magnitude. (**C**) Example of a set of four novel trials (HV and LV were kept

*Figure 1 continued on next page*

*Figure 1 continued*

constant across these four trials, but D varied). (D) Qualitative predictions of choice accuracy in the novel trials for the biophysical cortical attractor model proposed by Chau2014, the divisive normalization model proposed by Louie2013, various 'context effects' (see Materials and methods), and value-based attentional capture. The predictions vary with respect to the factors Similarity (i.e., whether D is more similar to HV or to LV) or Dominance (i.e., whether D is better or worse than HV/LV). In contrast to the other models, value-based attentional capture does not predict any influence of D's value on relative choice accuracy (grey bars) but a detrimental effect on absolute choice accuracy (white bars). Predictions of absolute choice accuracy for the remaining models are provided in *Figure 1—figure supplement 1*.

DOI: https://doi.org/10.7554/eLife.39659.003

The following figure supplement is available for figure 1:

**Figure supplement 1.** Qualitative predictions of absolute choice accuracy for alternative models.
DOI: https://doi.org/10.7554/eLife.39659.004

results was that the explicit presentation of the magnitude and probability attributes in the latter induced IIA-violating 'context effects' that emerge through multi-attribute comparison processes (*Gluth et al., 2017*; *Pettibone and Wedell, 2007*; *Roe et al., 2001*; *Trueblood et al., 2014*; *Usher and McClelland, 2004*; we elaborate on this and on further hypotheses in the Materials and methods). To test this hypothesis, we added a set of novel trials in which the high-value option (HV), the low-value option (LV), and the unavailable distractor (D) were positioned in the two-dimensional attribute space in a way to allow a rigorous discrimination between the various multi-attribute context effects, divisive normalization, and the biophysical cortical attractor model (*Figure 1C and D*). Note that henceforth, by 'divisive normalization' we refer to the model proposed by Louie2013 which assumes that normalization occurs at the level of integrated values (i.e., after combining attribute values such as magnitude and probabilities into a single value). Other models that instantiate hierarchical or attribute-wise normalization can make qualitatively different predictions (*Hunt et al., 2014*; *Landry and Webb, 2017*; *Soltani et al., 2012*). Also, we do not address the role of divisive normalization as a canonical neural computation (*Carandini and Heeger, 2011*) beyond its conceptualization by Louie2013.

Foreshadowing the results, our data from three behavioral and one eye-tracking experiments with a total of 147 participants are at stark odds with both the findings of Louie2013 and Chau2014. We did not observe a positive impact of D's value on relative choice accuracy as reported by Chau2014 in the same task, nor a negative impact of D's value as predicted by Louie2013. In other words, violations of IIA did not occur in the Chau2014 paradigm when participants made decisions under time pressure. Furthermore, a reanalysis of the original Chau2014 data suggested that the reported effect is a statistical artifact (see Materials and methods). In contrast to the absence of any effects on relative choice accuracy, however, we consistently found a negative impact of D's value on *absolute choice accuracy* (i.e., the overall probability of choosing the best option), which is not a violation of IIA. We argue that our behavioral and eye-tracking results as well as the results of the original study are best accounted for by *value-based attentional capture*, that is, by assuming that options capture attention proportional to their value. Value-based attentional capture is a comparatively novel concept in attention research, which has repeatedly been demonstrated in humans and non-human primates (*Anderson, 2016*; *Anderson et al., 2011*; *Grueschow et al., 2015*; *Le Pelley et al., 2016*; *Peck et al., 2009*; *Yasuda et al., 2012*). With respect to the Chau2014 task, it implies that a higher-value D draws more attention and thereby interferes with the choice process. To explain the results from our behavioral and eye-tracking experiments, we integrate the concept of value-based attentional capture into the well-established framework of evidence accumulation in decision making (*Bogacz et al., 2006*; *Gluth et al., 2012*; *Gluth et al., 2015*; *Gold and Shadlen, 2007*; *Heekeren et al., 2008*; *Smith and Ratcliff, 2004*). Our model predicts that only absolute but not relative choice accuracy will be affected by the value of the third option (i.e., no violation of IIA; *Figure 1D*). It provides a novel and cognitively plausible mechanism of the complex interplay of value and attention on multi-alternative decision making.

# Results

## Experiment 1: High-value distractors impair choice accuracy

Experiment 1 (with $n_1$ = 31 participants) aimed at comparing different model predictions with respect to the novel set of trials we introduced (*Figure 1B and C*) and at establishing the core finding of Chau2014 by replicating it. In the decision-making task as introduced by Chau2014 (*Figure 1A*), two available options of different expected value, HV and LV, are presented in two randomly selected quadrants of the screen and participants are asked to choose the option with the higher expected value. In half of the trials, the remaining two quadrants are left empty, in the other half, a third distractor option, D, is shown in one quadrant but indicated as unavailable after 0.1 s. Then, participants have another 1.5 s to make their choice. The central behavioral analysis by Chau2014 was a logistic regression of *relative choice accuracy* (i.e., whether HV or LV was chosen while excluding all trials with other responses such as choosing D or the empty quadrant, or being too slow) on the value difference between the two available options, HV-LV, the sum of their values, HV +LV, the value difference of HV and D, HV-D, the interaction of value differences, (HV-LV)×(HV-D), and whether a distractor was present or not, D present. Most importantly, the authors reported a significantly negative regression coefficient of HV-D indicating that higher values of D increased choice accuracy.

To test whether our results replicate the findings of Chau2014, we analyzed decisions made in the (non-novel) trials that were identical to those used by Chau2014. Note that the choice sets in Chau2014 were generated by sampling magnitude and probabilities for HV, LV, and D until HV-LV and HV-D shared less than 25% variance. The options' average expected values in the resulting trials were 5.13 for HV (*SD* = 3.08; min = 0.5; max = 10.5), 3.51 for LV (*SD* = 3.02; min = 0.25; max = 9), and 4.32 for D (*SD* = 2.11; min = 0.25; max = 9). Although the overall choice performance in our data was strikingly similar to the original data (Table S1 in *Supplementary file 1*), the negative effect of HV-D on relative choice accuracy could not be replicated. Instead, the average coefficient was positive but not significant (*t*(30) = 1.40, *p* = .171, Cohen's *d* = 0.25; left panel of *Figure 2A*; complete results of all regression analyses for all experiments are reported in Tables S2-S5). Interestingly, analyzing *absolute choice accuracy* (i.e., including all trials) resulted in a significant positive regression coefficient of HV-D (*t*(30) = 5.14, *p* < .001, *d* = 0.92; right panel of *Figure 2A*). This suggests that a higher value of D lowered choice accuracy when all trials were taken into account.

To better understand why D's value has this negative impact on the probability of choosing HV, we analyzed the different possibilities of making errors (i.e., choosing LV, choosing D, choosing the empty quadrant, and being too slow) by testing whether they were predicted by D's value. The value of D had a significant effect on the probability of choosing D (*t*(28) = 5.92, *p* < .001, *d* = 1.11) and also on being too slow (*t*(29) = 2.66, *p* = .013, *d* = 0.49), whereas the probability of choosing LV (*t*(30) = −0.47, *p* = .645, *d* = −0.08) or the empty quadrant (*t*(23) = 1.07, *p* = .294, *d* = 0.22) were unaffected (*Figure 2B*). Notably, higher values of D also slowed down response times (RT) for HV and LV choices (*t*(30) = 6.17, *p* < .001, *d* = 1.11; *Figure 2C*).

We then looked at choice accuracy in the novel trials that we added to differentiate between specific context effects and the models proposed by Louie2013 and Chau2014. Performance was higher in trials with Ds that were dominated by either HV or LV (*Figure 2C*), which is the opposite of what is predicted by the Chau2014 model (*Figure 1D*). This effect of Dominance was significant with respect to both relative choice accuracy (*F*(1,30) = 5.85; *p* = .022, $\eta_p^2$ = .16) and absolute choice accuracy (*F*(1,30) = 18.89; *p* < .001, $\eta_p^2$ = .39), the former being consistent with divisive normalization, the latter being consistent with value-based attentional capture (see *Figure 1D*; but note that–inconsistent with divisive normalization–we could not replicate the effect on relative choice accuracy in Experiments 2 and 3; Figure 5, Table S6-S7 for complete results). Overall, behavior in the novel trials supported the notion that higher-valued Ds impair multi-alternative decision making.

In summary, we could not replicate the results on relative choice accuracy and the IIA violations reported by Chau2014 but obtained opposite effects. At first glance, these results seem to support the divisive normalization account by Louie2013 as an appropriate mechanistic explanation. Note, however, that this account specifically predicts changes in relative choice accuracy, for which we found only weak evidence (i.e., no effect of HV-D in the regression, no effect of D's value on LV-errors). On the other hand, the effects on absolute choice accuracy were stronger. Accordingly, we

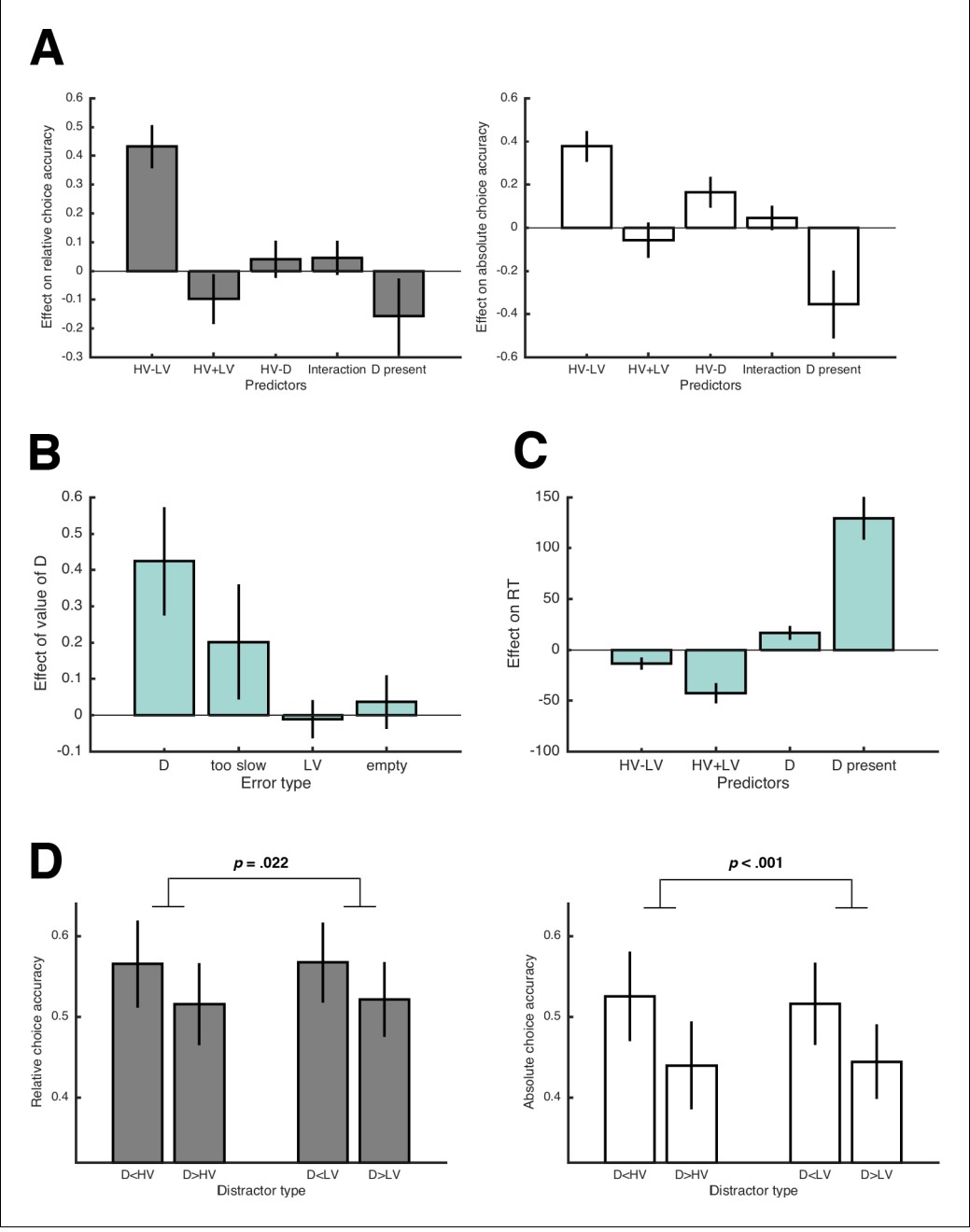

**Figure 2.** Results of Experiment 1. (**A**) Average regression coefficients of the predictor variables proposed by Chau2014 (including the HV-D predictor, which had a negative coefficient in their data) for relative and absolute choice accuracy (left and right panel, respectively). The term 'interaction' refers to the predictor variable (HV-LV)× (HV-D). Note that error bars in all figures represent 95% CIs, so that error bars not crossing the 0-line indicate significant effects. (**B**) Average coefficients of (separate) regression analyses testing how the value of D influenced different types of errors. (**C**) Average regression coefficients of predictor variables that influenced RT. (**D**) Relative and absolute choice accuracy in the novel trials (compare with predictions in *Figure 1D*). The comparatively low performance in the novel trials is due to HV and LV having very similar expected values (see *Figure 1C*).

DOI: https://doi.org/10.7554/eLife.39659.005

reasoned that the results are better explained by value-based attentional capture: This account states that attention is modulated by value such that value-laden stimuli attract attention and impair goal-directed actions, even if those stimuli are irrelevant to the task (*Anderson, 2016*; *Anderson et al., 2011*; *Grueschow et al., 2015*; *Le Pelley et al., 2016*; *Peck et al., 2009*; *Yasuda et al., 2012*). With respect to Chau2014's paradigm, value-based attentional capture predicts that high-value Ds draw attention to a greater extent than low-value Ds, which leaves less cognitive resources to focus on the decision between HV and LV. Consequently, decisions slow down, leading to higher RT and to more too-slow errors, and choices of D increase, but the relative probability of choosing HV over LV is unaffected–implying no violation of IIA. In Experiment 2 and 3, we sought to establish value-based attentional capture as the underlying mechanisms by testing different predictions that this explanation (but not divisive normalization) makes.

## Experiment 2: Distractor effects depend on decision time

If attentional capture is the driving force behind the performance decrease in the Chau2014 task, then increasing available attentional capacity should improve choice accuracy and reduce the detrimental effects of D. In contrast, divisive normalization effects do not seem to require the imposition of time pressure (see Louie2013). Based on this rationale, we conducted a second behavioral experiment in which we compared two groups: A high-time pressure (HP) Group ($n_{2,HP}$ = 25) did exactly the same task as in Experiment 1, but a second, low-time pressure (LP) Group ($n_{2,LP}$ = 24) was given more time to decide (6 s instead of 1.5 s). According to the attentional-capture account, we should replicate the results of Experiment one under high time pressure, but under low time pressure the negative influences of D on (absolute) choice accuracy should disappear. For the novel trial sets, we expected a choice accuracy pattern in line with multi-attribute context effects (see *Figure 1D*), given that these effects are known to become more prominent with longer deliberation time (*Dhar et al., 2000*; *Pettibone, 2012*; *Trueblood et al., 2014*).

Consistent with Experiment 1, we could not replicate the HV-D effect on relative choice accuracy reported in Chau2014. Again, our results showed a tendency in the opposite direction ($t(24)$ = 0.50, $p$ = .622, $d$ = 0.10). With respect to absolute choice accuracy, there was a strong and significantly positive effect of HV-D in Group HP of Experiment 2 ($t(24)$ = 5.05, $p$ < .001, $d$ = 1.01). In contrast and as predicted, this effect was much weaker and did not reach significance in Group LP ($t(23)$ = 1.85, $p$ = .079, $d$ = 0.38; but note that the difference between groups was not significant: $t(47)$ = 0.61, $p$ = .544, $d$ = 0.17; *Figure 3A*). Similarly, only participants of Group HP were less accurate in trials with D present compared to trials without D (Group HP: $t(24)$ = -5.97, $p$ < .001, $d$ = -1.19; Group LP: $t(23)$ = -0.06, $p$ = .956, $d$ = -0.01; group difference: $t(47)$ = -4.07, $p$ < .001, $d$ = -1.16; *Figure 3A*). Choices of D were significantly more frequent in Group HP (5.8% vs. 0.9% of trials; $t(47)$ = 4.15, $p$ < .001, $d$ = 1.20; *Figure 3B*). With respect to the novel trial set, we could not replicate the main effect of Dominance on relative choice accuracy in Group HP that we had found in Experiment 1 ($F(1,24)$ = 0.25; $p$ = .622, $\eta_p^2$= .01; left panel of *Figure 3C*). In Group LP, there was a significant main effect of Similarity ($F(1,23)$ = 9.20; $p$ = .011, $\eta_p^2$ = .25; right panel of *Figure 3C*) that was absent in any of our experiments conducted under high time pressure. This (IIA-violating) effect was due to a higher relative choice accuracy when D was more similar to HV than to LV in the two-dimensional attribute space and is consistent with a combination of an attraction effect (*Huber et al., 1982*) and a phantom-decoy effect (*Pettibone and Wedell, 2007*; *Pratkanis and Farquhar, 1992*; see *Figure 1D* and Materials and methods).

Taken together, while Group HP replicated the results of Experiment one in most aspects, Group LP showed that the negative impact of D disappeared when time pressure was alleviated, lending further support for a value-based attentional capture account. In addition, only Group LP exhibited an IIA-violating choice pattern in the novel trials consistent with specific context effects, supporting the notion that such effects require longer deliberation times to emerge (*Dhar et al., 2000*; *Pettibone, 2012*; *Trueblood et al., 2014*).

## Experiment 3: Value affects attention and attention affects decisions

Value-based attentional capture predicts an effect of the value of D on choice accuracy that is mediated by attention. In other words, a higher value of D leads to more attention to D, which in turn impairs absolute choice accuracy. This effect has been referred to as *value-based oculomotor*

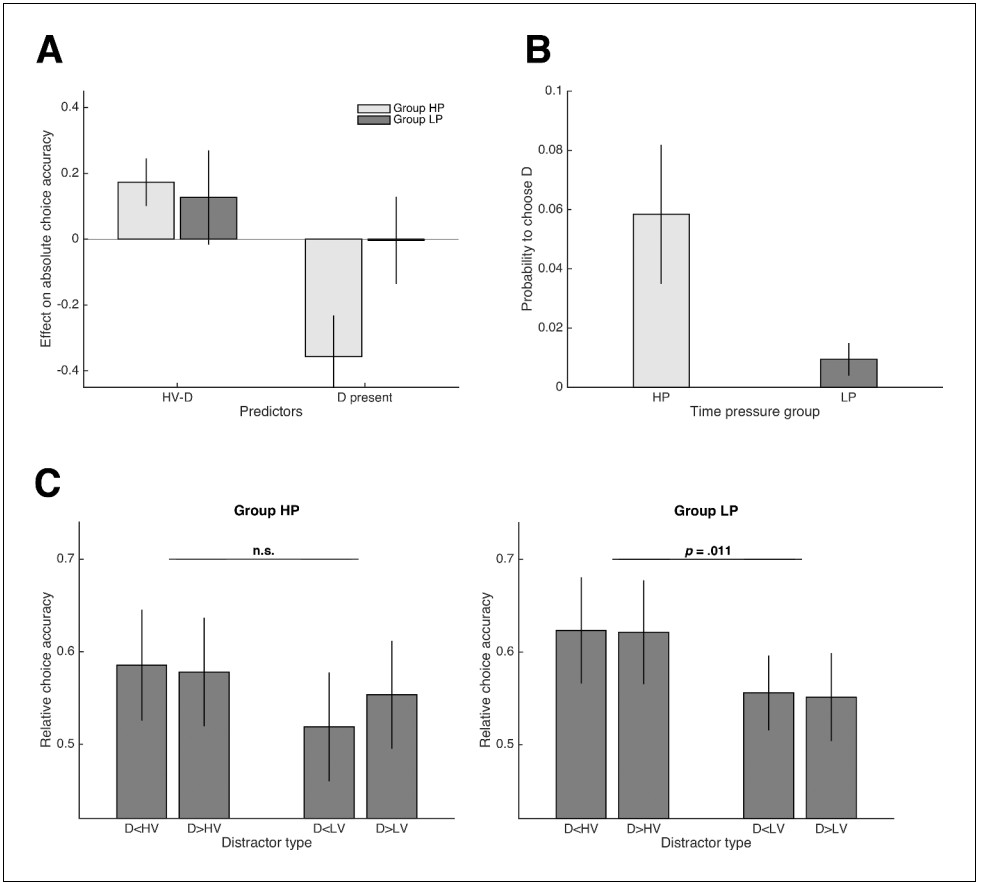

**Figure 3.** Results of Experiment 2. (**A**) Group comparison of regression coefficients reflecting the influence of HV-D and the presence of D on absolute choice accuracy (HP = high time pressure, LP = low time pressure). (**B**) Group comparison of the frequency of choosing D. In general, the negative influence of D on performance diminished substantially in Group LP, who made decisions under low time pressure. (**C**) Relative choice accuracy in the novel trials for Group HP (left panel) and LP (right panel). A violation of IIA was only observed in Group LP and is consistent with a combined attraction and phantom-decoy effect (compare with predictions in *Figure 1D*; see also Materials and methods).

DOI: https://doi.org/10.7554/eLife.39659.006

*capture* and is thought to underlie the (behavioral) effect of value-based attentional capture (*Failing et al., 2015*; *Le Pelley et al., 2015*; *Pearson et al., 2016*). To directly test whether value-based oculomotor capture drives the negative influence of D on decision making in the Chau2014 paradigm, we obtained eye-movement data as a measure of attention in Experiment three with $n_3$ = 23 participants using the same paradigm as in Experiments 1 and 2 (Group HP).

The behavioral results of this experiment were in line with our previous experiments (see Tables S2-S7 for statistical results): The negative HV-D effect on relative choice accuracy reported in Chau2014 could not be replicated, but its effect on absolute choice accuracy was significantly positive; the value of D led to more choices of D and slowed down choices of HV and LV; there was a main effect of Dominance on absolute (but not relative) choice accuracy in the novel trial set. With respect to the eye-movement results, we first tested whether options of higher value received more attention, defined as relative fixation duration (see Materials and methods). Thereto, we calculated within each participant the correlation between expected value and relative fixation duration separately for HV, LV, and D. These correlations were consistently higher than zero (HV: $t(22) = 4.79$, $p < .001$, $d = 1.00$; LV: $t(22) = 3.84$, $p < .001$, $d = 0.80$; D: $t(22) = 6.25$, $p < .001$, $d = 1.30$; *Figure 4A*). Second, we asked whether the dependency of attention to D on D's value mediated the negative influence of the latter on choice accuracy. A path analysis confirmed this prediction (*Figure 4B*): In the path model, the value of D had a positive effect on attention to D ($t(22) = 6.30$,

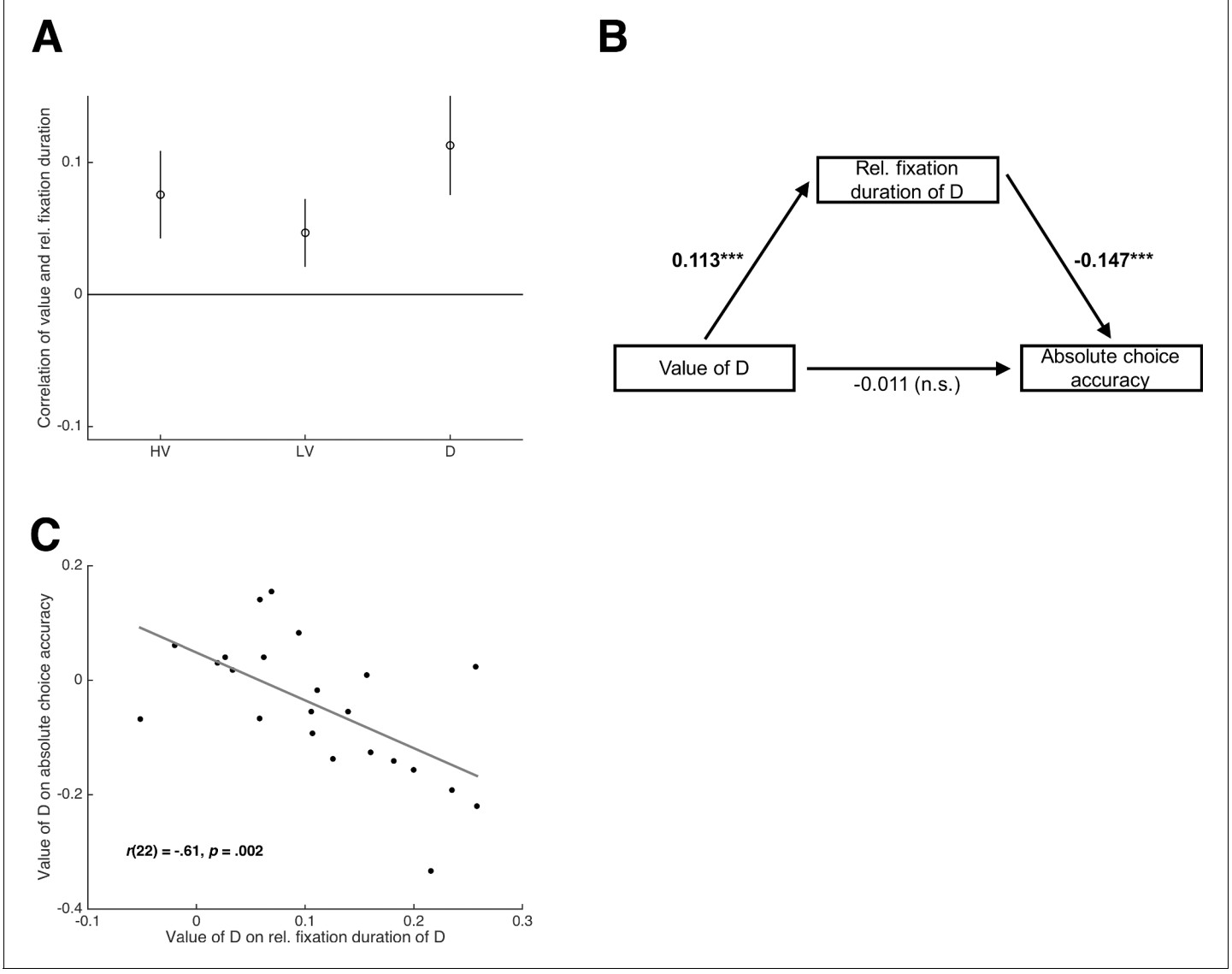

**Figure 4.** Eye-tracking results of Experiment 3. (**A**) Average coefficients for the correlation of the value of HV, LV, and D with the relative fixation duration on HV, LV, and D. (**B**) Path analysis of the relationship between the value of D, the attention on D (i.e., relative fixation duration of D), and absolute choice accuracy. The path analysis is conducted within each participant; the numbers represent the average path coefficients (tested against 0; ***$p < .001$). (**C**) Across-participant correlation of the coefficients representing the (positive) influence of the value of D on attention on D and the (negative) influence of the value of D on absolute choice accuracy.

DOI: https://doi.org/10.7554/eLife.39659.007

The following figure supplement is available for figure 4:

**Figure supplement 1.** Testing the influence of attention on choice.

DOI: https://doi.org/10.7554/eLife.39659.008

$p < .001$, $d = 1.31$), which in turn had a negative effect on choice accuracy ($t(22) = -6.79$, $p < .001$, $d = -1.42$). In fact, the influence of D's value on choice accuracy was fully mediated by attention with the direct path being not significant ($t(22) = -0.76$, $p = .455$, $d = -0.16$). Finally, we tested whether participants whose attention was affected by D's value to a greater extent also showed a stronger negative influence of D's value on absolute choice accuracy. This was confirmed by a significantly negative correlation across participants between the coefficient quantifying the correlation between D-value and D-fixations and the coefficient quantifying the negative impact of D-value on absolute choice accuracy ($r(21) = -0.61$; $p = .002$; **Figure 4C**). Altogether, the eye-movement results

strongly support value-based attentional and oculomotor capture as being the underlying mechanism of sub-optimality in the Chau2014 paradigm (see Materials and methods, *Figure 4—figure supplement 1* and *Figure 7—figure supplement 3* for further eye-tracking results).

## Experiment 4 and reanalysis of the Chau2014 dataset

We conducted a fourth experiment, for which we used the exact trial sequences of Chau2014 (provided to us by the authors) and omitted the novel trials. Besides replicating the effects of value-based attentional capture, another goal of this Experiment 4 was to test whether we could find the HV-D effect on relative choice accuracy when making the experiment almost identical to Chau2014. The statistical results of this experiment with $n_4 = 44$ participants are provided in Tables S2-S5 and are in line with Experiment 1 to 3: The negative HV-D effect on relative choice accuracy could not be replicated, there was a significantly positive HV-D effect on absolute choice accuracy, and higher values of D led to significantly more choices of D and to longer RT when choosing HV or LV. *Figure 5* summarizes the behavioral results collapsed over all experiments conducted under high time pressure. This summary provides clear evidence that there were no robust effects on relative choice accuracy (and thus no violations of IIA) but strong effects on absolute choice accuracy consistent with the value-based attentional capture account.

Notably, most of the behavioral data analyses in Chau2014 and our study relied on the assumption that participants can integrate magnitude and probability to compute the expected value (EV = probability × magnitude) of each option and to choose the option with the larger EV. We tested this assumption by comparing a simple EV-based choice model with two other models that assumed that participants focused on only one attribute (i.e., either magnitude or probability) as well as an expected utility (EU) model (e.g., *Von Neumann and Morgenstern, 1947*) and prospect theory (*Kahneman and Tversky, 1979*; *Tversky and Kahneman, 1992*); see Materials and methods). In brief, we found that the EV model explained the choice data better than the single-attribute models, but that the EU model provided the best account of the data (*Figure 5—figure supplement 1A*). The additional complexity of prospect theory compared to EU was not justified by an increased model fit. For reasons of simplicity and comparability to Chau2014, we used the EV-based value estimates for most statistical and modeling analyses. As a robustness check, however, we reanalyzed our main behavioral tests and replaced EVs by EU-based subjective values. The results were largely unaffected by this adaptation (*Figure 5—figure supplement 1B*).

We also reanalyzed the data of Chau2014 to look for further evidence of value-based attentional capture. As in our own experiments, we found that HV-D was positively linked to absolute choice accuracy ($t(20) = 4.53$, $p < .001$, $d = 0.99$), and that higher values of D led to both more erroneous choices of D ($t(19) = 4.67$, $p < .001$, $d = 1.05$) and to higher RT when choosing HV or LV ($t(20) = 2.94$, $p = .008$, $d = 0.64$). Thus, the data of Chau2014 also supported value-based attentional capture. In addition, we show with multiple additional analyses that are detailed in Materials and methods and in *Figure 5—figure supplement 2* to *Figure 5—figure supplement 5* that the originally reported negative effect of HV-D on relative choice accuracy is a statistical artifact that is due to an incorrect implementation of the interaction term (HV-LV)×(HV-D) in the performed regression analysis. After correcting this error, the effect of HV-D disappears.

## A computational model integrating value, attention, and decision making

Although value-based attentional capture is a well-established empirical finding (*Anderson, 2016*; *Le Pelley et al., 2016*), to the best of our knowledge it has never been implemented into a decision-making model so far. In the following, we propose and test the **M**utual **I**nhibition with **V**alue-based **A**ttentional **C**apture (**MIVAC**) model that accounts for the complex interplay of value and attention on choice (*Figure 6*; details are provided in Methods). MIVAC is an extended mutual inhibition (MI) model, a sequential sampling model that assumes a noisy race-to-bound mechanism of separate, leaky, and mutually inhibiting accumulators for each choice option (*Bogacz et al., 2006*; *Usher and McClelland, 2001*). Notably, the MI model (without the extensions we propose for MIVAC) is equivalent to a mean-field reduction of the cortical attractor model that was applied by Chau2014 (*Bogacz et al., 2006*; *Wang, 2002*; *Wong and Wang, 2006*), and it is indeed capable of predicting a positive effect of the value of D on relative choice accuracy (see *Figure 6—figure*

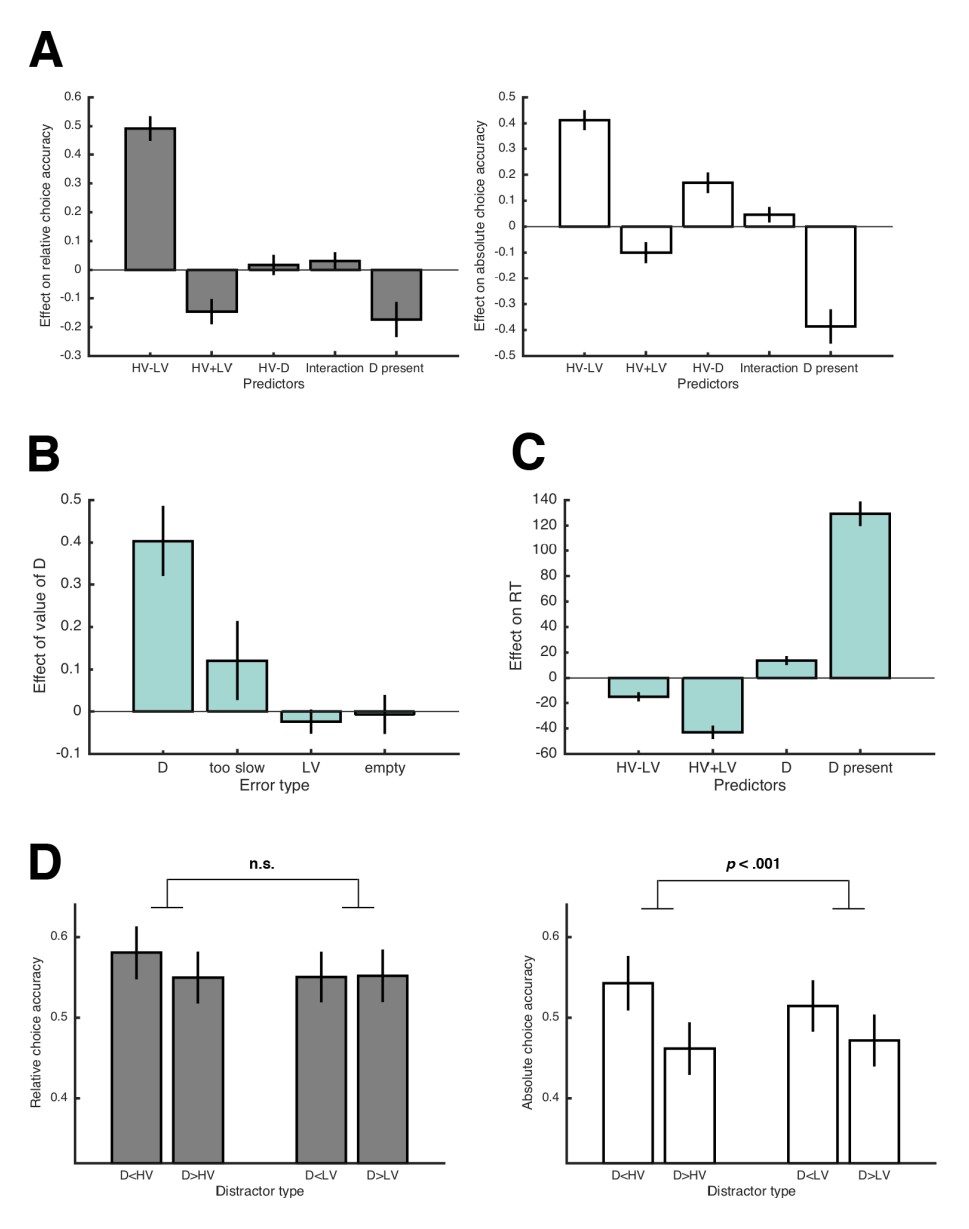

**Figure 5.** Summary of behavioral results of Experiment 1 to 4. (**A**) to (**D**) are analogous to *Figure 2* but collapsed over all experiments conducted under high time pressure. Results in (**D**) only contain data from Experiment 1 to 3, because novel trials were omitted in Experiment 4.

DOI: https://doi.org/10.7554/eLife.39659.009

The following figure supplements are available for figure 5:

**Figure supplement 1.** Testing the assumption that participants decided on the basis of expected values (EVs).

DOI: https://doi.org/10.7554/eLife.39659.010

**Figure supplement 2.** The influence of the interaction term (HV-LV)×(HV-D) on the estimation of the HV-D effect.

DOI: https://doi.org/10.7554/eLife.39659.011

**Figure supplement 3.** Effect sizes and test of detectability.

DOI: https://doi.org/10.7554/eLife.39659.012

**Figure supplement 4.** Analysis of the HV-D effect on relative choice accuracy for different levels of HV-LV.

DOI: https://doi.org/10.7554/eLife.39659.013

**Figure supplement 5.** Bayesian analyses of the HV-D Effect on relative and absolute choice accuracy.

DOI: https://doi.org/10.7554/eLife.39659.014

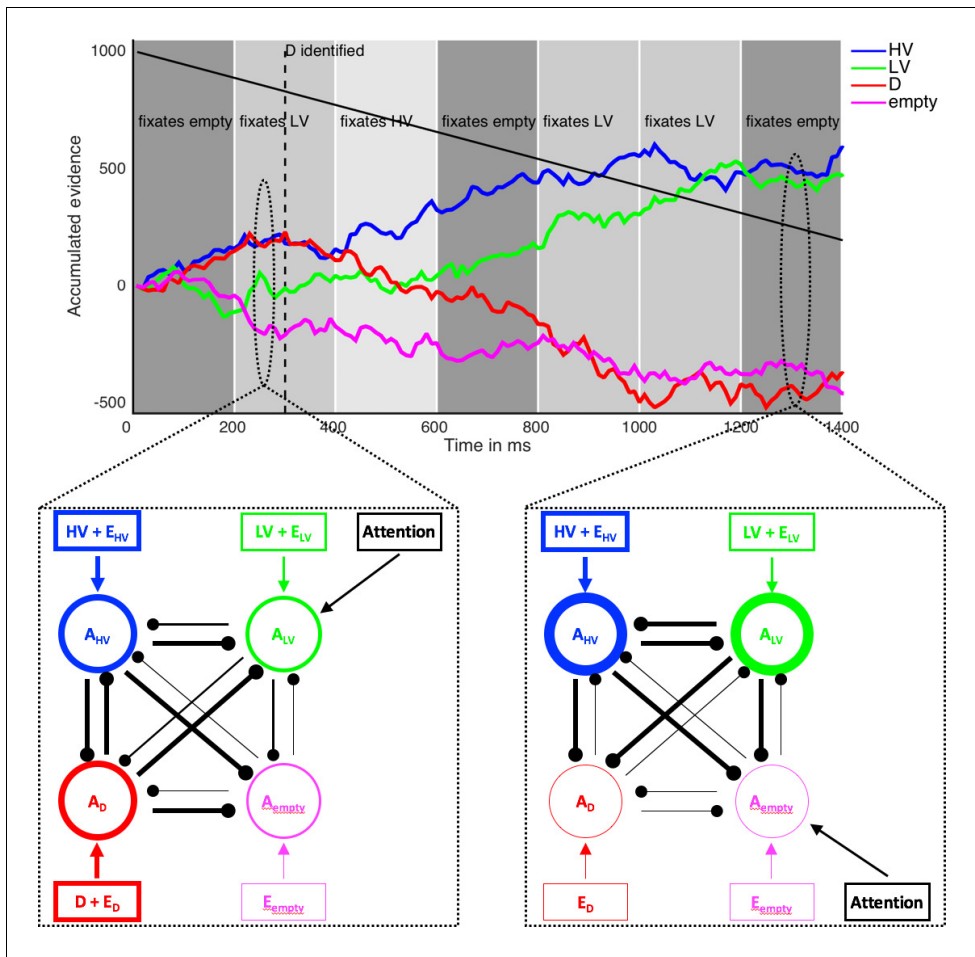

**Figure 6.** Illustration of the computational model MIVAC. Depicted is the development of accumulators of the MIVAC model with estimated parameters from a representative example participant in an example trial (upper panel) together with schematic outlines of the model at two different time points (lower panels). MIVAC consists of four accumulators representing HV, LV, D, and the empty quadrant. At every time step, each accumulator $A_x$ (round nodes in lower panels) receives an input (rectangular nodes) that is equal to the expected value of x (set to 7.5, 5, 10.5, and 0 in this example), plus Gaussian noise $E_x$. Accumulators inhibit each other (connecting lines between round nodes). Thicker round nodes, rectangular nodes/arrows, and lines indicate higher accumulation states, input, and inhibition, respectively. A choice is made as soon as an accumulator reaches an upper boundary that decreases with time (decreasing black line in upper panel). In this example, HV is chosen after ~900 ms. After D is identified (at 300 ms in this example; dashed vertical line in upper panel), the value of D does not serve as an input to its accumulator anymore. Every 200 ms, a new fixation is made (background greyscale), and the accumulator of the currently fixated option receives an additional input (black 'Attention' rectangular nodes). According to the value-based attentional capture element of MIVAC, the probability of fixating an option depends on its (relative) value.

DOI: https://doi.org/10.7554/eLife.39659.015

The following figure supplement is available for figure 6:

**Figure supplement 1.** The influence of D on relative choice accuracy in the MI and MIVAC models.
DOI: https://doi.org/10.7554/eLife.39659.016

---

*supplement 1*). MIVAC assumes an accumulator for each choice option, so four accumulators in the case of the Chau2014 paradigm (for HV, LV, D, and the empty quadrant). A choice is made as soon as an accumulator reaches an upper boundary that collapses in time (to account for the time limit of the task; for example *Gluth et al., 2012*; *Gluth et al., 2013a*; *Gluth et al., 2013b*; *Hutcherson et al., 2015*; *Murphy et al., 2016*). Based on our behavioral and eye-movement results, we propose three additional mechanisms. First and foremost, value-based attentional capture is

implemented by assuming that the probabilities of fixating particular options depend on their expected values. In other words, more valuable options receive more attention. Second, the input to the accumulator of the currently fixated option is enhanced, consistent with the influence of attention on choice reported in previous work (*Cavanagh et al., 2014*; *Krajbich and Rangel, 2011*; *Krajbich et al., 2010*; *Shimojo et al., 2003*) and observed in our own data (see *Figure 4—figure supplement 1* and Materials and methods). Finally, D can be identified as unavailable, in which case the expected value of D is assumed to be 0, and its accumulation rate and fixation probability are adjusted accordingly. This feature is a specific requirement for the Chau2014 paradigm in which participants are instructed not to choose D (this feature can be omitted for other experimental paradigms).

We propose MIVAC to explain the central behavioral findings across all experiments. Furthermore, we illustrate that simpler models without the added mechanisms of MIVAC (in particular, without value-based attentional capture) cannot explain these findings. Thereto, we conducted rigorous quantitative model comparisons in which MIVAC was compared to three models, each of them missing one of the three novel components (i.e., without VAC = without value-based attentional capture; without AE = without attentional enhancement; without DD = without distractor detection) and a baseline MI model, without any of these components. When fitting the models to trials in which a distractor D was present, we found a clear advantage of MIVAC compared to the simplified versions in terms of both average model fit and best model fit per participant (*Figure 7A*). We then performed a generalization test (*Busemeyer and Wang, 2000*) by using the models' estimated parameters from the trials with D to predict behavior in the trials without D. Again, MIVAC outperformed the simpler alternatives (*Figure 7B*). This generalization test provides strong support for a genuinely better description of the data by MIVAC and rules out an overfitting problem. Qualitatively, MIVAC predicts choice proportions of all potential actions very accurately, and it does so better than the simplified versions without value-based attentional capture with respect to both trials with D present and D absent (upper and middle panels of *Figure 7C*). The most important test for MIVAC was whether the model accurately predicts the observed increase in choices of D as a function of the value of D. As can be seen in the lower panel of *Figure 7C*, MIVAC predicts this pattern, and the alternative models either fail to do so or overpredict the frequency of choices of D. MIVAC also reproduces all RT effects reported in *Figure 5*, that is, negative effects of the difference and the sum of values of HV and LV, and positive effects of the value of D and the presence of D (*Figure 7D*; compare with *Figure 5C*). This is particularly remarkable because MIVAC was fitted only to the choice but not to the RT data. Finally, generalizing MIVAC to the novel trials shows that the model exhibits the observed qualitative patterns for both relative and absolute choice accuracy (*Figure 7E*; compare with *Figure 5D*). All these results hold when replacing the EV-based input to the accumulators by EU-based subjective values (*Figure 7—figure supplement 2*) or when simulating the model with fixations drawn from the empirical fixation duration distributions (*Figure 7—figure supplement 3*).

## Discussion

When choosing between multiple alternatives, humans and other animals often violate IIA, which has far-reaching consequences for our understanding of the neural and cognitive principles of decision making (*Hunt and Hayden, 2017*; *Rieskamp et al., 2006*; *Vlaev et al., 2011*). The purpose of our study was to shed light on the unresolved debate of whether the value of a third option leads to violations of IIA in the sense that it either decreases or increases relative choice accuracy between the other two options. Strikingly, we obtained strong evidence that neither violation of IIA is likely to occur when making decisions under time pressure, but that value-based attentional capture leads to a general performance decline (i.e., a decline in absolute but not relative choice accuracy representing no violation of IIA) and slows down the decision process. MIVAC, a computational model that we propose to explain the findings, is based on the assumptions that value drives attention (*Anderson et al., 2011*) and attention in turn affects the accumulation of evidence (*Krajbich et al., 2010*). We found that MIVAC reproduced the central behavioral findings for choice accuracy and RT with remarkable precision.

Specific characteristics of the Chau2014 task design are likely to have facilitated an influence of value-based attentional capture. Having to choose between four potential actions within about 1.5 s

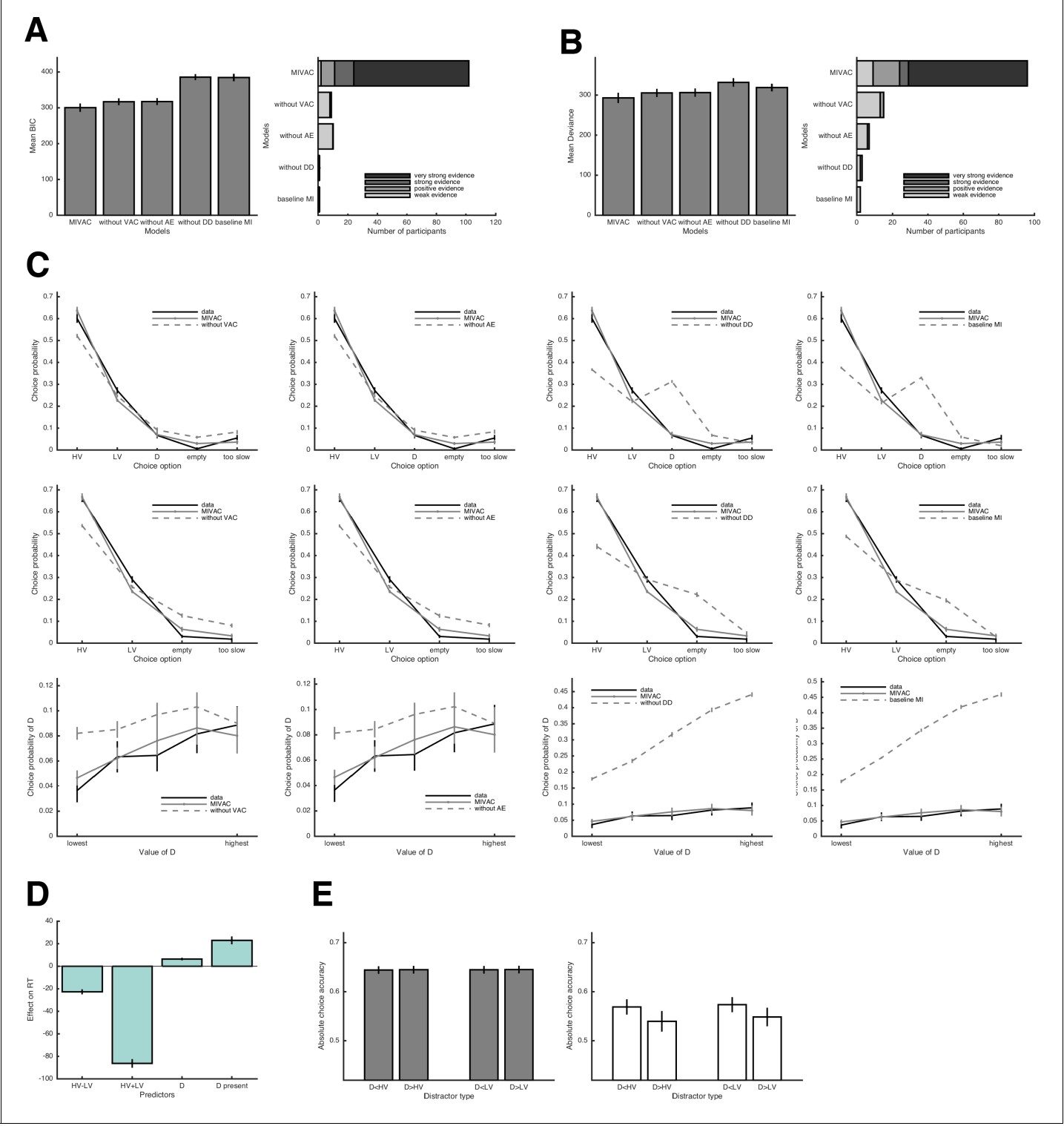

**Figure 7.** Quantitative and qualitative tests of MIVAC. (**A**) Left panel: average model fits for MIVAC and its simplifications for trials with D present (BIC = Bayesian Information Criterion; lower BICs indicate better fits). MIVAC had the lowest BIC ($p < .001$ for all pairwise comparisons). Right panel: model evidence per participant. (**B**) The same as in (**A**) but generalized to trials with D absent. MIVAC had the lowest deviance ($p < .001$ for all pairwise comparisons). (**C**) Upper panel: choice proportions (black line) for the five potential actions together with predictions of MIVAC (continuous grey line) and one of its variants (dashed grey lines). Middle panel: the same but for trials with D absent. Lower panel: observed and predicted probabilities of choosing D as a function of D's value. It can be seen that the variants of MIVAC without value-based attentional capture or attention-based enhancement of value accumulation fail to predict that choices of D increase as D's value increases. In contrast, the other two variants (i.e., without

*Figure 7 continued on next page*

*Figure 7 continued*

detection of D, baseline MI) predict too many choices of D. (D) Predicted RT effects of the value of D by MIVAC; the model correctly predicts all RT effects observed in the data (see *Figure 5C*). (E) Generalization of MIVAC to the novel trials; the model reproduces the observed qualitative patterns of both relative and absolute choice accuracy (see *Figure 5D*).

DOI: https://doi.org/10.7554/eLife.39659.017

The following figure supplements are available for figure 7:

**Figure supplement 1.** Model recovery results.

DOI: https://doi.org/10.7554/eLife.39659.018

**Figure supplement 2.** MIVAC with EU-based subjective values as inputs.

DOI: https://doi.org/10.7554/eLife.39659.019

**Figure supplement 3.** Fixation durations and MIVAC with empirical fixation durations.

DOI: https://doi.org/10.7554/eLife.39659.020

**Figure supplement 4.** Predictions of IIA violations with extension of MIVAC.

DOI: https://doi.org/10.7554/eLife.39659.021

**Figure supplement 5.** Comparison of MIVAC with multinomial logit (standard economic) choice models.

DOI: https://doi.org/10.7554/eLife.39659.022

puts participants under severe time pressure. This forces them to make use of implicit stimulus-reward associations for identifying attractive options as quickly as possible (*Krebs et al., 2011*; *Serences, 2008*). Using such implicit associations requires options that are distinguishable via low-level perceptual features, such as color or orientation (*Anderson, 2016*), which is exactly what was used in Chau2014. In fact, the Chau2014 paradigm closely resembles the visual search tasks used to study value-based attentional capture (*Anderson et al., 2011*), in which participants also have 1.5 s to identify a target out of several alternatives, while a distractor is characterized by a specific color (that was previously associated with low or high rewards). Yet, there are crucial differences between the two paradigms. Whereas the typical visual search tasks are framed as perceptual tasks, the Chau2014 paradigm is a value-based task. For example, choosing the LV option in the Chau2014 task can still lead to a reward, whereas choosing a non-target in the visual search task is treated as an error. Perceptual and preferential tasks have been shown to elicit different behavior (*Dutilh and Rieskamp, 2016*) and to rely on partially distinct neural mechanisms (*Polanía et al., 2014*). Such differences likely explain why we did not only find an influence of the value of D on RT but also on choice accuracy, which has been reported in only a minority of studies on value-based attentional capture (*Itthipuripat et al., 2015*; *Moher et al., 2015*).

An important role of attention in value-based decision making has been established in recent years (*Cavanagh et al., 2014*; *Cohen et al., 2017*; *Krajbich and Rangel, 2011*; *Krajbich et al., 2010*; *McGinty et al., 2016*; *Shimojo et al., 2003*). With some exceptions (*Itthipuripat et al., 2015*; *Towal et al., 2013*), research on the interaction of attention and choice has focused on the bias that the former exerts on the latter: Individuals are more likely to choose options that receive comparatively more attention. We replicated this effect in our eye-tracking experiment (*Figure 4—figure supplement 1*). However, several other eye-movement patterns in our data differed from previous findings. First, participants were more likely to look at more valuable options (*Figure 4A*), and this effect was significant even for the first fixation in a trial (Materials and methods). Second, the duration of first fixations was not shorter but longer compared to middle fixations (*Figure 7—figure supplement 1*). Third, the effect of attention on choice was not modulated by value (in the sense that the effect would be more pronounced for options of high compared to low value; *Figure 4—figure supplement 1*), which is in line with some (*Cavanagh et al., 2014*) but not with other previous findings (*Krajbich and Rangel, 2011*; *Krajbich et al., 2010*). Correspondingly, the MIVAC model differs from previous attention-based instantiations of sequential sampling models, in particular the attentional Drift Diffusion Model (aDDM; *Krajbich and Rangel, 2011*; *Krajbich et al., 2010*), in at least two aspects. First, attention is not distributed randomly across choice options but depends on the options' values (i.e., value-based attentional capture), and second, the accumulation of evidence for attended options is enhanced additively (i.e., independently of value). The latter feature is more in line with the results and model presented in *Cavanagh et al. (2014)*. Interestingly, this study also used abstract choice stimuli as compared to the food snacks used in the studies by Krajbich and colleagues, and participants were put under at least mild time pressure (i.e., choices had to be made

within 4 s). We conclude that attention and its influence on decision making can depend to some extent on the experimental design (*Spektor et al., 2018b*).

An alternative explanation for the interplay of attention, value, and choice could be that attention does not influence choice, but that choice intention influences attention. Stated differently, when having the intention to choose an option this option will receive more attention which then leads to longer fixation durations, so that high-value options are looked at longer only because they are more likely to be chosen. This reversed interpretation of the role of eye movements is particularly plausible for later stages of an ongoing decision because the last fixation is often directed at the eventually chosen option (*Shimojo et al., 2003*; but see *Krajbich et al. (2010)*, for an account of this 'gaze-cascade effect' within the aDDM framework). However, our analyses of first fixations and first-fixation durations are incompatible with such an interpretation (see Materials and methods). As stated above, the probability to fixate an option first depended on that option's value, consistent with value-based attentional capture. Importantly, this effect remained significant after controlling for the eventual choice, suggesting that it does not simply reflect the intention to choose the first-fixated option. Furthermore, we found that not only the first fixation and its duration but also the value of the first-fixated option contributed to predicting the eventual choice. In our view, this combined contribution of gaze time and value is best accounted for by assuming that people accumulate evidence for choosing an option based on both the option's value and the amount of attention spent on it. Therefore, we implemented these mechanisms into MIVAC accordingly.

At first glance, an effect opposite to what was observed by Chau2014 appears as support for the divisive normalization account by Louie2013. However, our results refute such an interpretation because the relative choice proportions between HV and LV were unaffected by the value of D. As with the different eye-movement results discussed above, the absence of a 'divisive normalization' effect in our data might be related to dissimilarities between the task paradigms. First of all, participants in Chau2014 (but not Louie2013) made decisions under time pressure, which promoted value-based attentional capture effects but suppressed violations of IIA. Furthermore, the decoy was highlighted as unavailable in Chau2014, whereas in Louie2013, the decoy was simply defined as the option with the lowest value. Finally, Chau2014 used abstract two-dimensional stimuli, whereas Louie2013 used concrete food snacks, for which it is currently debated whether and how specific attributes are taken into account (*Rangel, 2013*). Louie2013 did not address the distinction between single- and multi-attribute decisions, but their model should predict a negative influence of D's value on relative choice accuracy in both cases (because it assumes normalization to occur at the level of the integrated value signal). Although it goes beyond the scope of the current study, it will be critical to test to what extent effects of value-based attentional capture can be generalized to different implementations of multi-alternative decisions in future research.

Notably, the principle of normalization can be implemented into sequential sampling models, albeit in a different way compared to the mutual inhibition mechanism of MIVAC. According to Teodorescu and colleagues (2016), normalization acts on the input level of the accumulation process such that the input to one accumulator is reduced by the input to all other accumulators (a mechanism also referred to as 'feed-forward inhibition'; see *Bogacz et al., 2006*), whereas mutual inhibition acts on the accumulator level such that the accumulation of evidence for one option is reduced by how much evidence has been accumulated for the competing options. Separating between these different instantiations of inhibition is best achieved by taking their RT predictions into account (*Teodorescu et al., 2016*).

Generally, we did not obtain any reliable value-dependent violation of IIA in the standard version of the Chau2014 paradigm, but only when we gave participants more time to decide. In line with previous research (*Dhar et al., 2000*; *Pettibone, 2012*; *Trueblood et al., 2014*), these findings demonstrate the time-dependency of context effects: Being under time pressure or not determines whether effects related to value-based attentional capture or multi-attribute context effects can be expected to occur. One reason for this dependency could be that people adaptively select decision strategies based on the current decision context (*Gluth et al., 2014*; *Payne et al., 1988*; *Rieskamp and Otto, 2006*), and not all strategies are prone to the same contextual or attentional biases. Furthermore, multi-attribute context effects could be the consequence of dynamic choice mechanisms that require comparatively long deliberation times to exert a measurable influence on decisions (*Roe et al., 2001*; *Trueblood et al., 2014*).

Importantly, even though the negative influence of D on absolute choice accuracy is not a violation of IIA, our results cannot be well accounted for by standard economic choice models, such as the multinomial logit model (e.g., *McFadden, 2001*). We compared three variants of multinomial logit models with MIVAC (see Materials and methods and *Figure 7—figure supplement 5*). Two of these models either exclude or include D as a regular choice option. Excluding D leads to the prediction that D does not decrease absolute choice accuracy at all. Including D leads to the prediction that D is chosen as a function of its value relative to HV and LV and thus to the prediction that D is chosen in too many trials. A third alternative is to assume that choices are based on a combination of the options' EVs (as signaled by the rectangles' colors and orientations) with a subjective value assigned to the colors of the frames that signal whether an option is a target or a distractor. Although this version of a multinomial logit model is able to predict that D is chosen in only a minority of trials, its predictions are clearly worse than those of MIVAC, as the logit model predicts too few choices of HV and too many choices of LV (*Figure 7—figure supplement 5*). Furthermore, standard economic choice models do not take RT into account and thus cannot predict that high-value D options slow down the choice process (*Figure 5C*) such that the probability of making too-slow errors is increased (*Figure 5B*). A dynamic component, like the accumulation of evidence as implemented in MIVAC, is required to explain these RT-related effects. Also note that even though the current version of our model does not predict violations of IIA (consistent with our data under high time pressure), it is straightforward to extend MIVAC to allow such violations. In Materials and methods, we describe one possible extension of MIVAC and demonstrate how it allows the model to predict the IIA violation in our data under low time pressure.

Methodologically, the current study is an eminent example of the importance of replication attempts for the advancement of empirical science (*Munafò et al., 2017*), which is particularly timely given the current debate on replicability of research in psychology, neuroscience and other fields (*Camerer et al., 2016*; *Open Science Collaboration, 2015*; *Poldrack et al., 2017*). Our initial hypothesis for explaining the IIA violation reported in Chau2014 was based on the assumption of a robust and reliable effect. However, we could not replicate this effect (see Materials and methods and *Figure 5—figure supplement 2* to *Figure 5—figure supplement 5* for additional analyses that challenge both the replicability and the reproducibility of the effect proposed by Chau2014). When developing novel ideas and experiments on the basis of past findings, it is important that these findings are reliable and have been replicated, to avoid leading research fields into scientific cul-de-sacs, in which time and resources are wasted (*Munafò et al., 2017*). In our case, only the (unsuccessful) attempt to replicate an original study allowed us to identify truly robust behavioral effects, which favor an entirely different mechanistic explanation than originally proposed: Attention can be captured by irrelevant but value-laden stimuli, which slows down the goal-directed choice process and impairs decision making under time pressure. Crucially, neither our study nor replication studies in general are destructive. We gained novel insights about the role of attention in multi-alternative decision making, and we will gain similarly important insights when attempting to replicate other studies.

## Materials and methods

### Participants

Thirty-one participants (21 female, age: 20 – 47, *M* = 27.71, *SD* = 6.59, 29 right-handed) completed Experiment 1. A total of 51 participants signed up for Experiment 2. Due to computer crashes, the data of two participants (one from Group HP and one from Group LP) were incomplete and excluded from the analyses, resulting in a final sample of 49 participants, 25 were in Group HP (13 female, age: 20–46, *M* = 26.88, *SD* = 6.62, 24 right-handed) and 24 were in Group LP (11 female, age: 19–35, *M* = 23.96, *SD* = 3.43, 20 right-handed). Participants were assigned randomly to the two groups. Thirty participants signed up for the eye-tracking Experiment 3. One participant was excluded for not passing the training-phase criterion, and additional six participants were excluded due to incompatibility with the eye-tracking device (for further details see section *Eye-tracking procedures and pre-processing*), resulting in a final sample of 23 participants (14 female, age: 18–54, *M* = 25.70, *SD* = 8.66, 19 right-handed). Forty-seven participants signed up for Experiment 4. Due to failing the training-phase criterion, three participants were excluded, resulting in a final sample of

44 participants (36 female, age: 18–46, $M$ = 23.70, $SD$ = 5.74, 40 right-handed). The sample size for Experiment 1 was based on the assumption that testing 1.5 times as many participants as in the original study (Chau2014) would suffice to replicate its main results. In Experiment 4, we tested more participants to ensure a statistical power of >0.90 for replicating the effect of HV-D on relative choice accuracy (note that the effect size in Chau2014 was $d$ = −0.495). The sample size for Experiments 2 and 3 were based on observing strong effects (i.e., $d \geq 0.8$) of value-based attentional capture in Experiment 1 (but it should be noted that no formal power analysis was conducted). All participants gave written informed consent, and the study was approved by the Institutional Review Board of the Department of Psychology at the University of Basel. All experiments were performed in accordance with the relevant guidelines and regulations. Data of all participants included in the final samples are made publicly available on the Open Science Framework at https://osf.io/8r4fh/.

## Task paradigm

The paradigm was very similar for all four experiments. Participants repeatedly chose between either two (binary trials) or three (distractor trials) two-outcome lotteries (gambles), each yielding an outcome of magnitude X in Swiss Francs (CHF) with probability $p$ or 0 otherwise. The gambles were represented by rectangles, each shown in a random quadrant of the screen, whereby the rectangles' colors represented outcomes X and the angles represented the probabilities $p$. Outcomes ranged from CHF 2 to CHF 12 in steps of CHF 2 and were represented by colors ranging from either green to blue or blue to green. Probabilities ranged from 1/8 to 7/8 in steps of 1/8 and were represented by orientation angles ranging from 0° to 90° or from 90° to 0° in steps of 15° (see *Figure 1B* for all colors and orientations). Associations between colors/outcomes and orientations/probabilities were counter-balanced between participants. In binary trials, participants saw the two options for 100 ms before orange frames appeared around each option (pre-decision phase). After the frames appeared, participants had up to 1.5 s to make a choice by pressing 7, 9, 1, or 3 on the numeric keypad for upper left, upper right, lower left, or lower right quadrant, respectively (participants belonging to Group LP of Experiment 2 had 6 s after appearance of the frames to decide). Distractor trials were similar to binary trials: All of the options were presented for 100 ms before frames appeared around them. Contrary to the binary trials, one of the options had a magenta frame (the distractor), signaling that it could not be chosen. Choosing the distractor resulted in a screen telling that the option was not available after which a new trial began. Similarly, choosing an empty quadrant resulted in a screen showing that the quadrant was empty after which a new trial began.

If a valid choice was registered, the trial continued with a choice-feedback phase for 1 to 3 s, in which the chosen option was highlighted in a dark red color. To ensure that participants paid attention to all available options, participants had to complete a 'match' trial before the choice-feedback phase in 15% of all trials. On match trials, one of the options from the decision phase (including the distractor, if distractor trial) was presented in the middle of the screen. Participants had up to 2 s (6 s in Experiment 2, Group LP) to press the key corresponding to the option's quadrant. If correct, participants saw a screen saying 'correct' and an extra CHF 0.10 were added to the participant's account, otherwise they saw a screen saying 'wrong'. After every match trial, the trial continued with the choice-feedback phase.

After the choice-feedback phase, participants received feedback about the outcomes of the gambles for 1 to 3 s. The frames' colors changed to grey if the option did not yield a reward (i.e., the outcome was CHF 0) or to golden yellow if the option yielded a reward. Participants also received feedback about the distractor's outcome on distractor trials. After this outcome-feedback phase, a new trial began with an inter-trial interval of 1.5 to 3 s in which a fixation cross was shown.

## Experimental procedures

After giving informed consent and filling out the demographic questionnaire, participants received detailed instructions about the task and were familiarized with the outcome and probability associations by making six judgments for each dimension in a paired comparison. In all experiments, participants completed a training phase and an experimental phase. In the training phase, participants encountered up to 210 trials, half of which were distractor trials. These trials were randomly generated with the boundary condition that 2/3 of the trials were not dominant (i.e., HV did not have a higher probability *and* a higher outcome than LV) and the rest were dominant. The training phase

continued until participants encountered at least 10 dominant binary trials, and chose HV in at least 70% of the last 10 encountered dominant binary trials. The training phase ended when this criterion was reached and the experimental phase began. If the participants finished all 210 training trials without passing the criterion, the experiment ended. As reported above, participants who did not pass the criterion were excluded from the analysis.

The experimental phase consisted of either 412 (Experiment 1 to 3) or 300 (Experiment 4) trials. The 300 trials used in Experiment four were shared across all experiments and are those used by Chau2014. In Experiment 4, all trials were presented in exactly the same orders as in Chau2014's experiment, whereas in the other experiments, the distractor trials were presented in the order provided to us by the authors of Chau2014 with the randomized binary trials interleaved. In addition to these trials, Experiments 1 to 3 included 56 novel distractor trials and, correspondingly, the 56 binary trials belonging to these novel trials (details are provided in the next two sections). Throughout the experiment, participants had the opportunity to make four breaks. After completing the experiment, participants received their show-up fee (CHF 5 for 15 min), the average reward of the chosen option (distractor and empty quadrant choices counted as no reward), and the accumulated match bonuses. If participants reached the experimental phase, the experiments took approximately 75 min.

## Initial hypothesis for the effects reported in Chau2014

Before conducting our experiments, we assumed that the positive relationship between the value of D and relative choice accuracy as reported by Chau2014 was a robust effect. To explain the apparent contradiction with the findings of Louie2013, we reasoned that the explicit presentation of two attributes (i.e., magnitude X and probability $p$ of reward) in the Chau2014 task led people to compare the options on those attributes directly (i.e., a multi-attribute decision between attributes X and $p$ instead of a decision between expected values, EV). Importantly, certain attribute-wise comparison processes are known to produce IIA violations, so-called 'context effects of preferential choice' (*Berkowitsch et al., 2014*; *Gluth et al., 2017*; *Mohr et al., 2017*; *Pettibone and Wedell, 2007*; *Roe et al., 2001*; *Trueblood et al., 2014*; *Usher and McClelland, 2004*). More specifically, our initial hypothesis was that individuals can recognize that D is either better or worse than LV and/or HV with respect to each attribute (e.g., D might dominate LV with respect to probability). Critically, since LV is per definition worse than HV, it is more likely that D dominates LV than that it dominates HV on some attribute. However, this is only true as long as D has not very low attribute values (leading to a low EV overall). Thus, the *dominance relationship* between D and LV/HV may help participants to identify the option with the highest EV. The positive relationship between D and relative choice accuracy would then be an epiphenomenon of this dominance relationship mechanism.

To give an example, let us assume that HV, LV, and D are specified as follows:

HV: $p$ = 5/8; X = CHF 8 → EV = CHF 5
LV: $p$ = 3/8; X = CHF 4 → EV = CHF 1.5
D: $p$ = 6/8; X = CHF 6 → EV = CHF 4.5

In this case, D is superior to HV and LV with respect to probability. With respect to magnitude, however, D is superior to LV but inferior to HV. Hence, by counting the number of times D is superior to LV and HV on the attributes (i.e., two for LV vs. one for HV), a decision maker could correctly identify the option with the highest EV. Critically, this information is not helpful anymore when we replace D by a distractor D* of lower EV:

D*: $p$ = 6/8; X = CHF 2 → EV = CHF 1.5

In this new case, the distractor is inferior to both HV and LV with respect to magnitude and (still) superior to both with respect to probability (i.e., the count is one for LV vs. one for HV). Thus, the option with the highest EV cannot be identified anymore based on the dominance relationship alone. This demonstrates that the high-value D might better support choosing between HV and LV than the low-value D*. Notably, the idea that the relative ranking of options influences decision making has also been suggested by others (*Howes et al., 2016*; *Stewart et al., 2006*; *Tsetsos et al., 2016*).

Besides this dominance relationship hypothesis, the paradigm of Chau2014 might also involve other context effects that influence behavior. We considered the *attraction effect* (*Huber et al., 1982*) and the *phantom-decoy effect* (*Pettibone and Wedell, 2007*; *Pratkanis and Farquhar, 1992*). According to the attraction effect, the preference between two options (in our case between HV and LV) can be changed by adding a third option (in our case D) that is similar but clearly inferior

to only one of the two options (in our case, if D is similar to HV and worse than it, HV should be preferred; if D is similar to LV and worse than it, LV should be preferred). The phantom-decoy effect predicts that if an option is similar but worse than D (and D is unavailable, as in the Chau2014 task), it is more likely to be chosen. According to a combination of attraction and phantom-decoy effects, independent of whether D is worse or better than the similar option, the option being more similar to D should be more likely be chosen. In contrast to the context effects, the Chau2014 model, and the divisive normalization account, value-based attentional capture does not predict any influence of D on relative choice accuracy, but a negative effect on absolute choice accuracy (see *Figure 1D*). As described in the following section, we sought to distinguish between all these different context effects and the model proposed by Chau2014 by implementing a novel set of trials.

## The novel trial set and predictions of different models and context effects

In the novel set of trials used to dissociate predictions from various models and context effects, the HV and LV options were arranged such that i.) HV was superior to LV with respect to one attribute (magnitude or probability) but ii.) inferior with respect to the other attribute, and iii.) D could be placed such that it either fully dominated HV/LV or it was fully dominated by HV/LV (for an example, see *Figure 1C* in the main text). In the Chau2014 task, there are 14 possible combinations of HV and LV that fulfil these criteria. For each of these 14 combinations, D was placed directly 'above' or 'below' HV or LV resulting in four trials per combination and 56 novel trials in total (we also added 56 binary trials without D, so that Experiment 1 to 3 had 112 trials more than the original study).

The (qualitative) predictions of the models and context effects with respect to these novel trials are outlined in *Figure 1D*. Chau2014's biophysical cortical attractor model predicts a positive effect of the value of D on relative choice accuracy. Thus, the performance should be higher in trials with dominant distractors (i.e., D > HV and D > LV). The divisive normalization model by Louie2013 predicts higher accuracy for low-value Ds, or in other words, it predicts the opposite of Chau2014's model. Our initial dominance relationship hypothesis predicts higher choice accuracy if HV dominates D (i.e., D < HV), or if LV is dominated by D (i.e., D > LV), because in these cases HV is better than D on more attributes than LV. The combination of attraction and phantom-decoy effects predicts more accurate choices when D is dominated by HV (i.e., D < HV) or dominates HV (i.e., D > HV), or in other words, when D is more similar to HV than to LV. Importantly, all these predictions refer to relative choice accuracy (for which we do not find any robust effects in the Chau2014 task with short deliberation time; see *Figure 5D*). On the contrary, value-based attentional capture predicts no effect on relative choice accuracy, but a reduction of absolute choice accuracy when D has a high value (i.e., D > HV and D > LV).

Note that the different predictions can also be formulated in terms of main effects and interactions of an ANOVA with the factors Dominance (D dominates or is dominated by HV/LV) and Similarity (D is more similar to HV or to LV). Within this ANOVA, the cortical attractor model predicts a main effect of Dominance. The divisive normalization model also predicts this main effect but in the opposite direction. The dominance relationship hypothesis predicts an interaction effect of Dominance and Similarity. The combined attraction/phantom-decoy effect predicts a main effect of Similarity. Value-based attentional capture predicts a main effect of Dominance in the same direction as the divisive normalization model but on absolute (not relative) choice accuracy.

## Behavioral data analysis

In each trial, there was always a higher-valued option HV and a lower-valued option LV. We used two different dependent measures, *relative* and *absolute* choice accuracy. Relative choice accuracy refers to the proportion of HV choices among choices of HV and LV only, whereas absolute choice accuracy refers to the proportion of HV choices among all choices (including choices of D, choices of the empty quadrant, and missed responses due to the time limit). Importantly, an influence of the value of D on relative choice accuracy implies a violation of IIA, but an influence on absolute choice accuracy does not necessarily imply this. For each of the two dependent variables, we estimated intra-individual logistic regressions and tested the regression coefficients between subjects against 0 using a two-sided one-sample *t*-test with an α level of .05. We used the set of predictor variables reported in Chau2014, which consisted of the difference in EV of the two available options, HV-LV,

the sum of their EVs, HV+LV, the EV difference between HV and D, HV-D, the interaction between HV-LV and HV-D, (HV-LV)×(HV-D), and whether it was a binary or distractor trial, D present. In the binary trials, the predictors HV-D and (HV-LV)×(HV-D) were kept constant (i.e., replaced by the mean values in the distractor trials in the regression analysis). The predictor variables HV-LV, HV+LV, and HV-D were standardized. Importantly, standardization was conducted before generating the interaction term (HV-LV)×(HV-D) in order to avoid *nonessential* multicollinearity (*Aiken and West, 1991*; *Mahwah et al., 2003*; *Dunlap and Kemery, 1987*; *Marquardt, 1980*). The interaction term itself and the binary predictor D present were not standardized (note that this would not have changed any statistical inferences but only the absolute values of coefficients). In addition, we analyzed the influence of the (standardized) value of D on the tendency to choose D (logistic regression) and on the RT of HV and LV choices (linear regression). The RT analysis included additional predictor variables with a significant influence on RT (i.e., HV-LV, HV+LV, D present). Note that only the 300 trials that were also used in the Chau2014 paradigm (but not our 112 novel trials) were included in these regression analyses. The novel trial sets used to dissociate different model predictions were analyzed by a 2 (Dominance) x 2 (Similarity) ANOVA (for details see above).

Because participants received feedback after each decision, we tested whether improvements over time affected the results of relative or absolute choice accuracy by re-analyzing the regressions with an additional predictor variable that coded for the (standardized) trial number. Although we found a small learning effect on absolute choice accuracy ($t(122) = 2.12$, $p = .036$, $d = 0.19$), this did not affect any other effects qualitatively. When adding the trial number predictor to the analysis of the influence of D's value on the tendency to choose D, we found a strong learning effect ($t(122) = -10.18$, $p < .001$, $d = -0.92$), suggesting that participants improved in avoiding to choose D over the course of the experiment (but the effect of D's value remained significant). Hence, future instantiations of our model could incorporate a dynamic component to accommodate this learning effect.

## Eye-tracking procedures and pre-processing

Experiment 3 was conducted while participants' gaze positions were recorded using an SMI RED500 eye-tracking device. The experimental procedure was adapted to make it suitable for an eye-tracking experiment. Participants completed the experiment on a $47.38 \times 29.61$ cm screen (22" screen diagonal) with a resolution of $1680 \times 1050$ pixels. During the inter-trial interval, participants were instructed to look at the fixation cross and the random duration of the inter-trial interval was removed. Instead, there was a real-time circular area of interest (AOI) with a diameter of 200 pixels around the fixation cross. Participants' gazes had to (continuously) stay within this AOI for 1 s for the trial to begin. This was done to make sure that participants were indeed looking at the fixation cross and to check the calibration at every trial. If this criterion was not reached within 12 s, the eye tracker was re-calibrated. This procedure was explained to the participants by the experimenter. In case these re-calibrations happened too frequently (e.g., three times in a row within the same trial, or at least three times in ten trials), the sampling frequency was reduced from the initial 500 Hz to 250 Hz. If the issues continued until the lowest possible frequency of 60 Hz was reached, the experiment was aborted and participants received their show-up fee and decision-based bonuses accumulated until then. The first calibration took place just before the training phase and the eye tracker was re-calibrated after each of the four breaks. This experiment took approximately 90 min to complete.

The raw gaze positions were re-coded into events (fixations, saccades, and blinks) in SMI's BeGaze2 software package using the high-speed detection algorithm and default values. AOIs were defined around the positions where the frames of the options were, and all fixations inside the frame were counted towards the option within that quadrant. Fixations at empty quadrants as well as all fixations outside of the pre-defined AOIs were counted as *empty gazes*. Fixations within a trial were collapsed and summed to form the dependent variables *relative fixation duration* and *number of fixations*. We report results based on the relative fixation duration (i.e., the sum of the duration of all fixations on a specific quadrant divided by the sum of the duration of all fixations on any quadrant). Note that this measure is highly correlated with the number of fixations, which yielded similar results.

## Eye-tracking analysis I: Tests of value-based attentional and oculomotor capture

To test the hypothesis that the negative influence of D on absolute choice accuracy is driven by value-based attentional/oculomotor capture, we conducted the following three analyses: i.) We tested for the dependency of relative fixation duration of HV, LV, and D on their respective EVs (*Figure 4A*). Thereto, the correlations between the options' relative fixation durations and EVs were calculated for each participant, and the individual Fisher z-transformed correlation coefficients were subjected to one-sample *t*-test against 0 on the group level (after checking for normality assumptions via the Kolmogorov-Smirnoff test at $p < .1$). ii.) We conducted a path analysis (within each participant) in which the influence of the value of D on absolute choice accuracy was hypothesized to be mediated by the relative fixation duration on D (*Figure 4B*). iii.) We run an (across-participant) correlation between the behavioral regression coefficients representing the influence of the value of D on absolute choice accuracy and the regression coefficients representing the influence of the value of D on relative fixation duration on D (*Figure 4C*).

## Eye-tracking analysis II: Direct vs. value-dependent influences of attention on choice

To inform our computational model, we tested whether there is evidence for a direct effect of attention on choice (i.e., an option that receives relatively more attention is more likely to be chosen–independent of the option's value) as reported by Cavanagh and colleagues (*Cavanagh et al., 2014*), or whether the effect of attention is modulated by value (i.e., only options with high value that receive more attention are more likely to be chosen) as proposed by Krajbich and colleagues (*Krajbich and Rangel, 2011*; *Krajbich et al., 2010*). Thereto, we conducted a logistic regression analysis with absolute choice accuracy as dependent variable and the following six predictor variables: the expected value of HV, the expected value of LV, the relative fixation duration of HV, the relative fixation duration of LV, the interaction of the value of HV and the relative fixation duration of HV, and the interaction of the value of LV and the relative fixation duration of LV. The first three predictors were standardized, and the interaction variables were generated based on standardized variables to avoid *nonessential* multicollinearity (*Marquardt, 1980*). A *direct* attention model predicts a positive effect of the relative fixation duration of HV and a negative effect of the relative fixation duration of LV. A *value-dependent* attention model predicts positive and negative effects of the interaction terms for HV and LV, respectively. The results are shown in *Figure 4—figure supplement 1*. Besides the positive effect of HV-LV, there were significantly positive and negative effects of the relative fixation duration of HV ($t(22) = 9.40$, $p < .001$, $d = 1.96$) and LV ($t(22) = -8.20$, $p < .001$, $d = -1.71$), respectively. Of the interaction terms, only LV was significantly negative ($t(22) = -4.09$, $p < .001$, $d = 0.85$), but HV was not significantly positive ($t(22) = -1.21$, $p = .237$, $d = -0.25$). Thus, we found unequivocal evidence for a direct effect of attention on choice (*Cavanagh et al., 2014*) but not for an interaction of attention and value (*Krajbich and Rangel, 2011*; *Krajbich et al., 2010*). Consequently, we implemented the influence of attention on choice in the computational model by a value-independent, additive increase of the input signal to the accumulator of the currently attended option (see *Equation 9* below).

## Eye-tracking analysis III: Analysis of initial fixations

A central and often-tested prediction of the aDDM (*Krajbich and Rangel, 2011*; *Krajbich et al., 2010*) is that the direction of the initial fixation is independent of value (*Konovalov and Krajbich, 2016*; *Krajbich and Rangel, 2011*; *Krajbich et al., 2010*). However, the results of our eye-tracking experiment are inconsistent with this prediction. We found that the probability to fixate HV, LV, D first was positively linked to the respective option's value (HV: $t(22) = 6.03$, $p < .001$, $d = 1.26$; LV: $t(22) = 3.84$, $p < .001$, $d = 0.80$; D: $t(22) = 3.57$, $p = .002$, $d = 0.74$). Importantly, the effects for HV and D remained significant when controlling for the eventual choice (HV: $t(22) = 4.43$, $p < .001$, $d = 0.92$; LV: $t(22) = 0.62$, $p = .539$, $d = 0.13$; D: $t(21) = 2.70$, $p = .013$, $d = 0.58$), indicating that the effect was not solely driven by an intention to choose a particular option. Instead, the effect can be explained by assuming that value-based oculomotor capture biases early competition on the saccade map (*Pearson et al., 2016*).

In addition, we tested whether the first fixation and its duration was predictive of choice (when controlling for HV-LV and D), which would be consistent with previous findings (*Cavanagh et al., 2014*; *Krajbich and Rangel, 2011*; *Krajbich et al., 2010*) and suggest a biasing influence of attention on evidence accumulation. Indeed, the probability to choose HV, LV or D was predicted by whether the first fixation was made to the respective option (HV: $t(22) = 6.11$, $p < .001$, $d = 1.27$; LV: $t(22) = 4.98$, $p < .001$, $d = 1.04$; D: $t(20) = 7.02$, $p < .001$, $d = 1.53$). Similarly, the duration of the first fixation predicted the choice of all three options (HV: $t(22) = 7.26$, $p < .001$, $d = 1.51$; LV: $t(22) = 6.18$, $p < .001$, $d = 1.29$; D: $t(20) = 2.92$, $p = .008$, $d = 0.64$). Yet, the value difference HV-LV remained being predictive of the choice of HV and LV (HV: $t(22) = 8.09$, $p < .001$, $d = 1.69$; LV: $t(22) = -11.49$, $p < .001$, $d = -2.40$), and the value of D remained being predictive of the choice of D ($t(20) = 4.88$, $p = .003$, $d = 1.07$). Thus, the first fixation and its duration contributed to the probability of choosing an option but was not fully predictive of it. In our view, these results are best explained by assuming that gaze time biases the evidence accumulation towards the fixated option as implemented in the MIVAC model. Note that the analysis of first fixations included trials from the training phase.

## Testing the assumption of integration of magnitude and probability information

Many analyses in Chau2014 and our study, such as the regression analysis of relative choice accuracy, relied on the assumption that participants were able to integrate magnitude and probability information to calculate the expected value (EV) of each option and to choose the option with the largest EV. We tested this assumption by comparing five simple choice models with respect to how well they predicted whether participants chose HV or LV (note that in contrast to MIVAC, we do not propose any of these models to account for all behavioral findings but instead treat them only as auxiliary models to test the assumption that participants relied on EV when making decisions). The first two models assumed that participants relied *only* on *magnitude* information ('OM') or *only* on *probability* information ('OP'). The third model assumed that participants relied on *EV* ('EV'). The fourth model assumed that participants relied on subjective values given by *expected utility* theory (e.g. *Von Neumann and Morgenstern, 1947*) ('EU'), specified as follows:

$$SV_{EU} = EU(x,p) = u(x) \times p \tag{1}$$

$$u(x) = x^{\alpha} \tag{2}$$

where $\alpha$ is a free parameter ($\alpha > 0$) modulating the curvature of the utility function. The last model assumed that participants relied on subjective values given by *prospect theory* (Kahneman and Tversky, 1979; Tversky and Kahneman, 1992) ("PT"), specified as follows:

$$SV_{PT} = V(x,p) = v(x) \times w(p) \tag{3}$$

$$v(x) = x^{\alpha} \tag{4}$$

$$w(p) = \frac{p^{\tau}}{(p^{\tau} + [1-p]^{\tau})^{\frac{1}{\tau}}} \tag{5}$$

where $\alpha$ is again a free parameter ($\alpha > 0$) modulating the curvature of the utility function and where $\tau$ is an additional free parameter ($\tau > 0$) modulating the shape of the probability-weighting function. For each of these models, the probability of choosing HV was given by the logistic/soft-max choice function:

$$P_{HV} = \frac{1}{1 + e^{-\delta \times \left(SV[HV]_{Model} - SV[LV]_{Model}\right)}} \tag{6}$$

where $\delta$ is a free parameter ($\delta > 0$) modulating the sensitivity to value differences, and $SV[HV]_{Model}$ and $SV[LV]_{Model}$ refer to the model-specific values of options HV and LV, respectively. We used the same minimization algorithm and model-comparison approach as for MIVAC (see section *Modeling*

*procedures for MIVAC III: parameter estimation and nested model comparison*). Models were compared on the basis of the choice data from the 300 trials that were part of the initial choice set of Chau2014 and from the 123 participants of our study who conducted the Chau2014 paradigm under time pressure. Because the EU model provided the most parsimonious account of the data (*Figure 5—figure supplement 1*), we reanalyzed the central behavioral tests of our study (shown in *Figure 5A–5C*) and replaced the options' EVs by the EU-based subjective values estimates. Furthermore, we reran the MIVAC model to test whether using EU-based subjective values as input to the model's accumulators would lead to qualitatively different predictions (*Figure 7—figure supplement 2*; details are provided in section *Modeling procedures for MIVAC IV: MIVAC with subjective values as inputs*). Note that the analyses and predictions for the novel trial sets (*Figures 1D* and *5D*) do not depend on whether EV or EU is assumed as the underlying choice model.

## Modeling procedures for MIVAC I: The baseline mutual inhibition model

The computational model we propose, MIVAC, is an extended version of the mutual inhibition (MI) model (*Bogacz et al., 2006*; *Usher and McClelland, 2001*). The MI model belongs to the sub-class of accumulator models within the framework of sequential sampling models (*Ratcliff and Smith, 2004*; *Smith and Ratcliff, 2004*). Sequential sampling models assume that evidence for choice options is accumulated over time until a decision threshold is reached after which a choice is made. Accumulator models use separate evidence accumulators for each choice option. MIs further assume mutual (or lateral) inhibition of these accumulators, that is, the more evidence for one choice option has been accumulated, the stronger this accumulator inhibits the competing accumulators (in contrast to feedforward inhibition models that assume inhibition between the inputs to accumulators). In addition, accumulators are leaky, that is, they tend to decay toward the starting point (i.e., 0). One reason why we chose this framework (and not, for instance, the aDDM) is that the MI is equivalent to a mean-field reduction of the biophysical attractor model that was applied by Chau2014 (*Bogacz et al., 2006*; *Wang, 2002*; *Wong and Wang, 2006*). Accordingly, without a value-based attentional capture mechanism, the MI model is able to predict a positive effect of the value of the distractor (D) on choice accuracy similar to the model proposed by Chau2014 (see *Figure 6—figure supplement 1*). In general, we begin with a *baseline MI* and then add additional assumptions (about the interplay of value, attention, and choice) that are necessary to explain the observed behavioral findings.

Formally, the MI assumes four accumulators representing the four choice options high-value option (HV), low-value option (LV), D, and the empty quadrant. The *n by t* matrix **A** representing all *n* (=4) accumulators is updated every time step $\Delta t$ according to:

$$\mathbf{A}_{t+1} = \mathbf{S} \cdot \mathbf{A}_t + \mathbf{I}_t + \mathbf{E}_t \tag{7}$$

where **S** represents an *n by n* leakage and inhibition matrix with leakage parameter $\delta$ and inhibition parameter $\phi$ as on- and off-diagonal elements, respectively, $\mathbf{I}_t$ represents an *n by t* input matrix that–in the case of the baseline MI–contains the EV of each choice option (which is 0 for the empty quadrant), and $\mathbf{E}_t$ represents an *n by t* noise matrix with each element being drawn from a normal distribution with mean 0 and standard deviation $\sigma$. As soon as accumulator $A_{i,t}$ reaches an upper threshold $\theta_t$ (i.e., $A_{i,t} \geq \theta_t$), option *i* is chosen (at time point *t*). Similar to previous work (*Gluth et al., 2012*; *Gluth et al., 2013a*; *Gluth et al., 2013b*; *Hutcherson et al., 2015*; *Murphy et al., 2016*), we assume that the decision threshold decreases (linearly) with time (note that we found the empirical response-time [RT] distribution in this task to lack the typical right-skewed shape, which is possibly due to the strict time limit and would be in line with a decreasing decision threshold; see *Hawkins et al., 2015*). In contrast to the leaky competing accumulator model (*Usher and McClelland, 2001*) but in line with suggestions of *Bogacz et al., 2006*, we allowed **A** to take on negative values.

Potentially, the baseline MI could have six free parameters: the leakage parameter $\delta$, the inhibition parameter $\phi$, the standard deviation $\sigma$ of the noise vector, the height of the decision threshold at the start and end of the decision period $\theta_0$ and $\theta_{tmax}$, and a non-decision time parameter $\tau_{non-dec}$. However, because we sought to extend this model by three additional components and because predictions of the MI cannot be derived analytically but must be simulated, we drastically constrained the model by fixing all but one of the aforementioned parameters (it should be noted at

this point that we had to make several simplifying assumptions to simulate the model in a reasonable amount of time). Leakage and inhibition were set to $\delta = 0.96$ and $\phi = -0.036$, respectively, the start and end points of the decision threshold were set to $\theta_0 = 1'000$ and $\theta_{tmax} = 200$, respectively, and the non-decision time was set to $\tau_{non\text{-}dec} = 200$ ms (so that $tmax - \tau_{non\text{-}dec} = 1'400$ ms). Thus, only $\sigma$ remained as a free parameter (this parameter controls the stochasticity of the decision process and allows to account for performance differences between participants). Importantly, fixed parameter values were chosen carefully so that the model made various sensible predictions (i.e., the relative choice ratio between HV and LV, the average RT, and the proportion of misses due to no threshold crossings until *tmax* were ensured to lie all within the range of the empirical observations).

## Modeling procedures for MIVAC II: Extending the baseline MI to MIVAC

For the MIVAC model, we propose three additional mechanisms that are required to account for the behavioral data and that describe the interplay of value, attention, and decision making as observed in the eye-tracking experiment. First, we implemented value-based attentional capture by modelling fixations to the choice options and by assuming that the probability of fixating a particular option depends on the option's value (relative to the other options). Similar to a recent study on reinforcement learning and attention (*Leong et al., 2017*), we used a logistic function to describe the relationship of value and fixation probability:

$$F_i = \frac{e^{\gamma \times nEV(i)}}{\sum_j e^{\gamma \times nEV(j)}} \tag{8}$$

where $F_i$ is the probability of fixating option *i*, $\gamma$ is a free parameter, and $nEV_i$ is the expected value of option *i* normalized with respect to all currently presented options (normalization was used to avoid that value-based attentional capture effects depend too much on the sum of all values). We assumed that every 200 ms, a new fixation is made, and with probability $F_i$ it is made to option *i* (with the possibility to re-fixate the same option). This time window roughly corresponds to the average fixation duration of 230 ms in our data. We chose this implementation to maintain a high speed of model simulations and to make more conservative model predictions, as a more data-driven implementation of using empirical fixation patterns (e.g., *Cohen et al., 2017*; *Krajbich and Rangel, 2011*; *Krajbich et al., 2010*) may improve model fit but limit generalizability. In the section *Modeling procedures for MIVAC V: MIVAC with fixation durations drawn from empirical distributions*, we show that using empirical fixation durations (instead of a fixed duration of 200 ms) does not change the models' predictions and the model comparison results qualitatively.

The second additional mechanism is the enhanced input to the currently attended accumulator. This assumption is based on previous work showing that options which receive more attention are more likely to be chosen (*Cavanagh et al., 2014*; *Krajbich and Rangel, 2011*; *Krajbich et al., 2010*; *Shimojo et al., 2003*), which we replicated in our eye-tracking experiment. Note, however, that we found evidence for a direct effect of attention on choice rather than for a modulatory effect that depends on the value of the attended option (see section *Eye-tracking analysis II: direct vs. value-dependent influences of attention on choice* and *Figure 4—figure supplement 1*). Accordingly, we assumed that the input $I_{a,t}$ of the currently attended option *a* is enhanced additively rather than multiplicatively:

$$I_{a,t} = EV(a) + \beta \tag{9}$$

where EV(*a*) is the expected value of *a*, and $\beta$ is a free parameter representing the attention-based enhancement of accumulation. Together with value-based attentional capture, this means that MIVAC assumes that options of higher value receive more attention, and that the evidence for an attended option is accumulated faster (but MIVAC does not assume that there is an interaction of value and attention, in the sense that the increase of accumulation due to attention is stronger for higher- compared to lower-valued options).

Finally, we assumed that participants can detect that the distractor is unavailable but occasionally fail to do so. This assumption was necessary because participants usually followed the instructions of the Chau2014 paradigm and avoided choosing D in most of the trials but still picked it more often than the empty quadrant: In trials with D present, there were 6.7% choices of D and only 0.1%

choices of the empty quadrant ($t$(122) = 12.17; p<0.001); in trials without D, there were only 3.1% choices of both empty quadrants together. Therefore, we assumed that after every 100 ms, D can be identified as being unavailable with probability π, which is a free parameter (restricting the identification of D to every 100 ms was again necessary to maintain fast simulations of the model; note that 100 ms also corresponds to the point in time at which participants were told which of the options D is). If D is identified, the input to its accumulator is set to 0 (plus noise) just as for the empty quadrant. The probability to fixate D is also affected by its identification (i.e., $nEV_D$ is set to 0 in *Equation 8*).

Taken together, MIVAC has four free parameters, the standard deviation σ of accumulation from the baseline MI model, the value-based attentional capture parameter γ, the attention-based enhancement β of accumulation, and the probability π to identify D as being unavailable (see Table S8 in *Supplementary file 1* for estimated parameter values). In addition to the baseline MI and MIVAC, we tested three simplified versions of MIVAC that each lacked one of the additional features (i.e., either γ, β, π, or all of them were set to 0).

It should be noted that the proposed model does not predict violations of IIA, such as the attraction or the phantom-decoy effect (for which we found evidence in the experiment with long deliberation time). We refrained from implementing this because our central goal was to model the Chau2014 paradigm with short deliberation time, for which we did not find any IIA violations, and because we wanted to maintain a high speed of model simulations. Importantly, however, MIVAC can be extended to allow predicting IIA violations straightforwardly, for instance by implementing a stage of attribute-wise comparisons that modify the input $I_t$ (*Gluth et al., 2017*; *Hunt et al., 2014*; *Roe et al., 2001*; *Trueblood et al., 2014*; *Usher and McClelland, 2004*). In the section *Modeling procedures for MIVAC VI: Extending MIVAC to allow for violations of IIA*, we provide one example of how such an extension could work, and show that it allows accounting for the IIA violation observed in the experiment with long deliberation time. Thus, in principle MIVAC is compatible with the rich literature on violations of IIA in value-based and perceptual decision making (*Berkowitsch et al., 2014*; *Gluth et al., 2017*; *Huber et al., 1982*; *Mohr et al., 2017*; *Rieskamp et al., 2006*; *Spektor et al., 2018a*; *Spektor et al., 2018b*; *Trueblood et al., 2013*; *Tsetsos et al., 2012*).

## Modeling procedures for MIVAC III: Parameter estimation and nested model comparison

As mentioned, parameter estimation required time-consuming simulations of the models. Similar to our previous work (*Gluth et al., 2013b*), we chose a step-size Δ$t$ of 10 ms for fast simulations, and approximated the likelihood by simulating every trial 100 times. Trial-wise predictions were truncated to a minimum probability of .01 and a maximum probability of .99 (*Gluth et al., 2013b*; *Nassar and Frank, 2016*; *Rieskamp, 2008*). Note that the estimation of parameters was based on predicting choices but not RT, because our main goal was to explain the choice data, and because predicting RT would have required more simulations per trial and estimating the non-decision time (and possibly also the decision threshold), which would have prolonged computations to an infeasible amount of time. As shown in *Figure 7D*, however, the model reproduces all of the observed RT effects (i.e., higher value differences and sums of HV and LV decrease RT, higher values of D and the presence of D increase RT). Parameter estimation was realized by combining an initial grid search algorithm to obtain reasonable starting values, which were then passed on to a constrained simplex minimization algorithm (*Nelder and Mead, 1965*) as implemented MATLAB's *fminsearchcon* to obtain the final estimates. For the grid search, 31 steps per parameter were used, equally spaced within the following bounds: σ between 5 and 50, γ between −1.12 and 3.08, β between −2.1 and 18.9, π between 0 and 1 (the same values were used as constrains during the simplex minimization). The simplex algorithm was run 16 times in total, 3 times with the parameter set obtained from the best grid search model fit, 9 to 13 times with a parameter set randomly selected from the best 1% of grid search model fits (20% in case of the baseline MI), and 0 to 4 times with the best parameter set obtained from the simplex search of nested models (e.g., all other models are nested in MIVAC, so the algorithm for MIVAC was started 4 times with the four other models' best parameters as starting values). The final estimates of parameters were taken from the simplex search with the best model fit. Only trials of the Chau2014 set with D present were used to estimate parameters. This

allowed testing a generalization (*Busemeyer and Wang, 2000*) of the models to the trials without D (*Figure 7B*) and to the novel trials that we added in Experiment 1 to 3 (*Figure 7E*).

The central goal of computational modeling was to compare MIVAC against simpler versions that lacked either one or all of the novel mechanisms. Thus, model comparison comprised the baseline MI, MIVAC, and three versions of MIVAC without either $\gamma$, $\beta$, or $\pi$ (which were set to 0 for the respective model). Since all other models are nested within MIVAC, they could be compared using a likelihood ratio test (*Lewandowsky and Farrell, 2011*). However, we used the Bayesian Information Criterion (BIC; *Schwarz, 1978*) to compare the models, because it is more conservative when comparing the most complex model against simpler ones, and because the BIC allows to estimate participant-wise model evidence: For each participant, we classified model evidence as 'weak', 'positive', 'strong', or 'very strong', when the BIC difference $x$ between the best and the second-best model was $x < 2$, $2 < x < 6$, $6 < x < 10$, or $x > 10$, respectively (*Raftery, 1995*). Following our previous work (*Gluth et al., 2017*), we tested the generalizability of the models by comparing their deviances for the trials without D and for the novel trials, which were not used for parameter estimation. We also compared the models on a qualitative level with respect to i.) how well they predict choices in D present trials, ii.) how well their predictions can be generalized to trials without D, and iii.) how well they account for the dependency of choosing D on the value of D (*Figure 7C*). Note, that all quantitative and qualitative model comparisons are based on out-of-sample predictions by running 10'000 new simulations per trial, participant, and model using the estimated parameters. A simple parameter recovery analysis (in which we generated synthetic data using MIVAC's estimated parameters of each participant and the 150 trials from the Chau2014 paradigm and re-estimated parameters for these synthetic data) confirmed that the model is identifiable with the set of trials used in Chau2014 and our study (for all parameters, correlations between data-generating and re-estimated parameter values were significantly positive at p<0.001; see *Figure 7—figure supplement 1*).

## Modeling procedures for MIVAC IV: MIVAC with subjective values as inputs

Similar to our behavioral analyses, MIVAC assumes that people are able to integrate magnitude and probability information to estimate an option's EV (since the EV serves as the basic input signal to each accumulator; see *Equation 9*). Because our analyses of the choice data suggested that an EU-based choice model provided a more accurate account of the data than an EV-based model (see *Figure 5—figure supplement 1*), we reran MIVAC and replaced the EVs as inputs to the options' accumulators by the EU-based subjective value estimates. Because the range of EU-based values heavily depend on the power utility parameter $\alpha$ (if $0 < \alpha < 1$, the range is narrower than the range of EVs; if $\alpha > 1$, the range is wider than the range of EVs), we ensured that subjective values were kept in the same range as the EVs by the following transformation (see *Berkowitsch et al., 2015*):

$$EU_{transformed} = min_{EV} + \frac{(EU - min_{EU}) * (max_{EV} - min_{EV})}{max_{EU} - min_{EU}} \tag{10}$$

We simulated responses from all models 10'000 times per trial and participant with the previously estimated parameters. As can be seen from *Figure 7—figure supplement 2*, the predictions of the EU-based MIVAC model are very similar to the EV-based MIVAC model.

## Modeling procedures for MIVAC V: MIVAC with fixation durations drawn from empirical distributions

As stated in the main text, we did not draw fixations from empirical fixation distributions (as has been done, for instance, by *Krajbich et al., 2010*) when estimating parameters of MIVAC for two reasons. First, we sought to avoid overfitting MIVAC by equipping it with actual fixation patterns: If the main behavioral effects had been solely due to differences in fixation patterns across options, then using those patterns would have allowed MIVAC to make accurate predictions, even though the model's proposed features (e.g., value-based attentional capture) would not have been critical. Second, without using empirical fixation distributions the model can be faster simulated and thereby fitted to the data in a feasible amount of time.

Here, we show that using empirical fixation patterns does not alter the predictions of MIVAC and its simplified competitor models qualitatively. Thereto, we first analyzed mean fixation durations by

splitting fixations according to the factors *Option Type* (i.e., HV, LV, D, empty quadrant) and *Fixation Type* (i.e., first vs. middle fixations; see *Krajbich et al., 2010*; *Figure 7—figure supplement 3A*). A $4 \times 2$ repeated-measures ANOVA revealed significant main effects for both *Option Type* ($F(3,18)$ = 29.51, $p < .001$, $\eta_p^2 = .83$) and *Fixation Type* ($F(1,20)$ = 25.70, $p < .001$, $\eta_p^2 = .56$) and a significant interaction ($F(3,18)$ = 18.24, $p < .001$, $\eta_p^2 = .75$). Accordingly, we approximated the empirical fixation-duration distributions by estimating the means and standard deviations of log-normal distributions (*Krajbich et al., 2010*) for each option and fixation type separately (*Figure 7—figure supplement 3B*). Finally, we simulated responses from all models 10'000 times per trial and participant with the previously estimated parameters. We sampled each fixation duration from the log-normal distribution with the respective means and standard deviations per option and fixation type (e.g., when the first fixation in a trial was on HV, then the fixation duration was drawn from a log-normal distribution with its mean and standard deviation taken from the fit to the HV/first distribution). As can be seen from *Figure 7—figure supplement 3C*, the predictions of MIVAC do not change qualitatively when empirical fixation durations are used. Moreover, using these fixation durations does not allow the simplified competitor models to catch up with MIVAC (*Figure 7—figure supplement 3D*).

## Modeling procedures for MIVAC VI: Extending MIVAC to allow for violations of IIA

As stated in the main text, MIVAC does not predict violations of IIA but can be extended straightforwardly to do so. Here, we provide one example of such an extension and show that it allows to predict the IIA violation we obtained in the experiment with long deliberation time (Experiment 2, Group LP). For the extension, we followed the assumption made by many multi-attribute sequential sampling models, including the Multialternative Decision Field Theory (*Roe et al., 2001*), the Leaky Competing Accumulator model (*Usher and McClelland, 2004*), and the Multi-attribute Linear Ballistic Accumulator model (MLBA; *Trueblood et al., 2014*), namely that the input to the accumulator of an option *i* is based on comparisons of this option with all other options. In our notation, the input of option *i* thus becomes

$$I_{i,t} = \sum_j V_{i,j} + I_0 + \beta \tag{11a}$$

if *i* is currently fixated,

$$I_{i,t} = \sum_j V_{i,j} + I_0 \tag{11b}$$

if *i* is currently not fixated,
where $I_0$ is a free parameter of the model ($I_0 \geq 0$), representing the baseline input to each accumulator, and $V_{i,j}$ represents the output of a comparison process between option *i* and option *j*. In specifying $V_{i,j}$, we follow the MLBA, according to which this comparison process is the weighted sum of the attribute-wise differences:

$$V_{i,j} = w_{i,j}^M * \left( u_i^M - u_j^M \right) + w_{i,j}^P * \left( u_i^P - u_j^P \right) \tag{12}$$

where is $w_{i,j}{}^M$ is the weight given to attribute *M* (for 'magnitude') in the comparison between options *i* and *j*, and $u_i{}^P$ is the subjective representation of attribute *P* (for 'probability') for option *i*. In the full MLBA, the subjective representations can be subject to non-linear transformations. For the sake of simplicity, however, we used the objective magnitudes and probabilities as $u^M$ and $u^P$, respectively. The weight given to an attribute *X* is a function of the similarity between the options with respect to that attribute *X*:

$$w_{i,j}^X = e^{-\lambda_1 * \left| u_i^X - u_j^X \right|} \tag{13a}$$

if $u_i{}^X \geq u_j{}^X$,

$$w_{i,j}^X = e^{-\lambda_2 * |u_i^X - u_j^X|} \tag{13b}$$

if $u_i{}^X < u_j{}^X$,

where $\lambda_1$ and $\lambda_2$ are two free parameters of the model that modulate the decay of attribute weights when the distance between the options' attribute values increases. Note that this means that perceived similarity is not symmetrical: A can be perceived as more similar to B than B to A. As stated above, the outputs from this attribute-comparison layer are the inputs $I_t$ to the accumulation process (*Equation 9*).

For our simulations, we set $I_0$ to 20, $\lambda_1$ to 3, and $\lambda_2$ to 0.05. While the value for $I_0$ was chosen simply to avoid negative inputs to accumulators, we chose to set $\lambda_1 > \lambda_2$ because then the weight $w_{i,j}{}^X$ of attribute X in the comparison between option i against option j will be higher when i is better than j with respect to X compared to when it is worse, with this difference being largest at close distances (because of the decay function that links $\lambda$ to $w$). As a consequence, option i will receive an enhanced input in the presence of a similar, inferior alternative, but it will not receive a reduced input in the presence of a similar, superior alternative. Such a disproportionate attention weight is a default assumption of MLBA and predicts both the attraction and the phantom-decoy effect, that is, an increased relative choice accuracy when (an inferior or superior) D is more similar to HV than to LV. All other parameter values were taken from the fit of the non-extended MIVAC model. The deliberation time was set to 6'000 ms. We ran 10'000 simulations per participant. As shown in *Figure 7—figure supplement 4*, the extended MIVAC is able to predict the pattern of IIA violation that we observed in Experiment 2 Group LP, namely that the probability of choosing HV is increased in the presence of a similar (inferior or superior) D option.

## Modeling procedures for MIVAC VII: Comparison of MIVAC with multinomial logit models

We also compared MIVAC to three different multinomial logit models as representatives of standard economic choice models (e.g., *McFadden, 2001*); see *Figure 7—figure supplement 5*). The first logit model ('Exclude D') assumes that participants are perfectly able to identify that D should not be chosen. Thus, irrespective of whether D is present or absent, the probability of choosing HV becomes a logistic function of the expected value (EV) difference between HV and LV:

$$P_{HV} = \frac{1}{1 + e^{-\delta \times (EV(HV) - EV(LV))}} \tag{14}$$

where $\delta$ is a free parameter reflecting the participant's sensitivity to EV differences. The second logit model ('Include D') assumes that participants are completely unable to identify that D should not be chosen. Thus, the probability of choosing an option i is:

$$P_i = \frac{e^{\delta \times EV(i)}}{\sum_j e^{\delta \times EV(j)}} \tag{15}$$

with D being part of the choice set and where the sum of EVs is taken for all choice options. The third logit model ('EV and Frame Value') assumes that D is treated as a regular choice option (as in the 'Include D' model), and that choices are based on an additive combination of the options' EVs with additional subjective values that are assigned to the differently colored frames (which indicate whether an option is a target or a distractor; see *Figure 1A*). Formally, the overall subjective value (SV) of an option i is:

$$SV(i)_{EV \text{ and } Frame\ Value} = EV(i) + FV(i) \tag{16}$$

where *FV(i)* is the 'Frame Value' of option i. We assumed that the Frame Value for distractors is 0 and the Frame Value for targets is a free parameter (restricted to positive values). The choice probability is then given by *Equation 15* with EV replaced by SV.

Maximum likelihood estimates of the models' parameter(s) were computed on the basis of trials with D present, using the same minimization algorithm as for MIVAC. Because the three models and MIVAC used different sets of data (i.e., the 'Exclude D' model used only trials in which either HV or LV were chosen, the other two logit models also included trials in which D was chosen, MIVAC

further included trials in which the empty quadrant was chosen or no decision was made), the models could not be compared via quantitative criteria. Instead, we compared their ability to predict the observed choice proportions of HV, LV, and D in trials with D present, and of HV and LV in trials with D absent (*Figure 7—figure supplement 5*).

## Reanalysis of the Chau2014 dataset

The authors of Chau2014 provided us with the behavioral data of their fMRI experiment that included 21 participants. We reanalyzed the data to check for evidence of effects related to value-based attentional capture: We tested for an effect of HV-D on absolute choice accuracy, for an effect of D's value on the propensity to choose D, and for an effect of D's value on RT. The results of all of these analyses were in line with predictions of a value-based attentional capture account (see Results).

When trying to reproduce the results of the original study, we also found that the HV-D effect on relative choice accuracy (i.e., the central behavioral result of Chau2014) was only significantly negative when the interaction term (HV-LV)×(HV-D) in the same regression analysis was generated on the basis of uncentered HV-LV and HV-D variables. In multiple regression analyses, centering (i.e., subtracting each entry by the mean) or standardizing (i.e., subtracting the mean and dividing by the standard deviation) predictors before generating their interaction term is critical to avoid *nonessential* multicollinearity (*Aiken and West, 1991*; *Mahwah et al., 2003*; *Dunlap and Kemery, 1987*; *Marquardt, 1980*). This is particularly important when one or both of the predictors do not have a meaningful 0 (i.e., are always positive or always negative). This is the case for HV-LV which is always positive by definition. Accordingly, the uncentered interaction term (HV-LV)×(HV-D) is correlated substantially with HV-LV ($r = .287$) and correlates particularly strong with HV-D ($r = .862$) (upper row of *Figure 5—figure supplement 2*). In contrast, the mean-centered interaction term is only weakly correlated with HV-LV ($r = .004$) and HV-D ($r = -.064$) (lower row of *Figure 5—figure supplement 2*).

To test how susceptible the original results of HV-D are to the generation and implementation of the interaction term, we conducted several control analyses. First of all, we repeated the regression analysis on relative choice accuracy, but used the correct interaction term (i.e., the term generated on the basis of standardized HV-LV and HV-D predictors). Strikingly (but in line with our own results), the effect of HV-D in the Chau2014 data was positive and not significant ($t(20) = 1.56$, $p = .134$; see also *Figure 5—figure supplement 3*, which shows the effect sizes of HV-D in all experiments when the correct vs. incorrect interaction term is used in the analysis). Second, we took the interaction term out of the regression analysis, which yielded similar results ($t(20) = 0.67$, $p = .510$). Third, we reanalyzed the original data in the following way: We separated the dataset by each level of HV-LV, which yielded 12 bins, and conducted the regression analysis on relative choice accuracy for each bin separately. This procedure allowed us to take out the interaction term (together with the predictor variable HV-LV itself), and to test for a 'cleaner' effect of HV-D in each bin. As shown in *Figure 5—figure supplement 4* a significantly negative effect of HV-D on relative choice accuracy is seen in only 1 of the 12 bins, but there is also a significantly positive effect in another bin. Conducting this analysis with our own data yielded very similar results. Fourth, we simulated data from hypothetical decision makers who decide solely on the basis of HV-LV (via a logistic/soft-max choice function [see *Equation 14* above] for which we set the choice-sensitivity parameter to 0.5 to roughly match the choice accuracy observed in the data). For these simulated participants, we know that their behavior is not influenced by HV-D. We conducted 1'000 simulations of 21 participants (the sample size of Chau2014) and performed four logistic regression analyses, either with or without the interaction term, and either with standardized or uncentered HV-LV and HV-D predictors. We found a spurious negative influence of HV-D on relative choice accuracy when including an interaction term with uncentered predictors ($t(999) = -3.43$, $p < .001$; upper row of *Figure 5—figure supplement 2*). This influence was absent when including the interaction term with standardized predictors ($t(999) = 0.36$, $p = .717$) or when excluding the interaction term with either standardized or uncentered predictors (both: $t(999) = -0.89$, $p = .375$; lower row of *Figure 5—figure supplement 2*). Taken together, the negative effect of HV-D on relative choice accuracy is a statistical artifact of the incorrect implementation of the interaction term (HV-LV)×(HV-D) into the logistic regression analysis.

## Test of detectability and Bayesian analysis

The reanalysis of Chau2014 raises strong doubts on the reproducibility of the original results. Yet, theoretically there might still be a true negative HV-D effect that in principle could have been found in our own four experiments (while being absent in the original study due to its low statistical power). To test for the 'replicability' of this effect without reverting to the (equivocal) effect size of the original study, we conducted a recently proposed *test of detectability* that is independent of the reported effect size of the original study (*Simonsohn, 2015*). This approach uses only the sample size (and the statistical design) of the original study to specify the (hypothetical) effect size that would have given the study only 33% statistical power, which can be regarded as undoubtedly insufficient. If the effect size of the study that attempts to reproduce the effect of the original study is significantly below this $d_{33\%}$ threshold, then one can conclude that the studied effect is not large enough to have been detectable with the original sample size. The results of this test for the HV-D effect on relative choice accuracy show that the 95% confidence intervals of the effect sizes for each of our Experiments 1, 2 HP, 3, and four as well for all these experiments combined and for the original study itself (when using the correct implementation of the interaction term) were indeed closer to 0 than the $d_{33\%}$ threshold for the sample size of 21 participants used in Chau2014 (*Figure 5—figure supplement 3*). But even when using the incorrect implementation of the interaction term (that biases the results toward more negative HV-D effects), the confidence intervals of the effect size for our four experiments combined are closer to 0 than the $d_{33\%}$ threshold.

Finally, we conducted a Bayesian analysis of the HV-D effects on relative (and absolute) choice accuracy to be able to quantify the evidence in favor of the null hypothesis (*Wagenmakers, 2007*). Specifically, we applied the *ttestBF* function of the *R* package *BayesFactor* (with default settings) to obtain Bayes Factors (BF) in favor of the null (or alternative) hypothesis that the HV-D regression coefficients of relative and absolute choice accuracy are equal to (or deviate from) 0. We included the 123 participants from Experiments 1, 2 HP, 3, and 4 who performed the task under high time pressure. For the HV-D effect on relative choice accuracy we obtained a $BF_{01}$ in favor of the null hypothesis of 20.05 when testing against the alternative hypothesis of a negative HV-D effect (*Figure 5—figure supplement 5*, upper panel). This means the data are about twenty times more likely to have occurred under the null hypothesis, which is also seen as *strong* evidence for the null hypothesis (*Kass and Raftery, 1995*). When testing against the alternative hypothesis of a positive HV-D effect (which would be the opposite of what Chau2014 had reported and in line with the divisive normalization account of *Louie et al., 2013*), we obtained a $BF_{01}$ in favor of the null hypothesis of 3.14, which can be considered as *positive* evidence for the null hypothesis (*Kass and Raftery, 1995*). For the HV-D effect on absolute choice accuracy we obtained a $BF_{10}$ in favor of the alternative hypothesis of $4.07 \times 10^{13}$, which is seen as *decisive* evidence (*Figure 5—figure supplement 5*, lower panel).

## Acknowledgments

We thank Bolton Chau and Matthew Rushworth for providing us with the data from the original study and for helpful and constructive discussions. We thank Chris Donkin for pointing us towards literature on value-based oculomotor capture.

## Additional information

### Funding

| Funder | Grant reference number | Author |
| --- | --- | --- |
| Schweizerischer Nationalfonds zur Förderung der Wissenschaftlichen Forschung | 100014_153616 | Sebastian Gluth<br>Jörg Rieskamp |

The funders had no role in study design, data collection and interpretation, or the decision to submit the work for publication.

## Author contributions
Sebastian Gluth, Conceptualization, Formal analysis, Supervision, Funding acquisition, Visualization, Methodology, Writing—original draft, Writing—review and editing; Mikhail S Spektor, Conceptualization, Formal analysis, Investigation, Visualization, Methodology, Writing—original draft, Writing—review and editing; Jörg Rieskamp, Conceptualization, Resources, Supervision, Funding acquisition, Methodology, Project administration, Writing—review and editing

## Author ORCIDs
Sebastian Gluth ⬥ http://orcid.org/0000-0003-2241-5103
Mikhail S Spektor ⬥ https://orcid.org/0000-0003-0652-1993

## Ethics
Human subjects: All participants gave written informed consent, and the study was approved by the Institutional Review Board of the Department of Psychology at the University of Basel. All experiments were performed in accordance with the relevant guidelines and regulations.

## Decision letter and Author response
Decision letter https://doi.org/10.7554/eLife.39659.028
Author response https://doi.org/10.7554/eLife.39659.029

# Additional files

### Supplementary files
• Supplementary file 1. Tables for statistical results and model parameters
DOI: https://doi.org/10.7554/eLife.39659.023

• Transparent reporting form
DOI: https://doi.org/10.7554/eLife.39659.024

### Data availability
Behavioral and eye-tracking data have been deposited on the Open Science Framework at https://osf.io/8r4fh/

The following dataset was generated:

| Author(s) | Year | Dataset title | Dataset URL | Database and Identifier |
|---|---|---|---|---|
| Mikhail S Spektor, Sebastian Gluth, Jörg Rieskamp | 2017 | Value-based attentional capture affects multi-alternative decision making | https://osf.io/8r4fh/ | Open Science Framework, 8r4fh |

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
