## [Decision Letter]

Thank you for submitting your article "Value-based attentional capture affects multi-alternative decision making" for consideration by *eLife*. Your article has been reviewed by two peer reviewers, and the evaluation has been overseen by Michael Frank as the Senior and Reviewing Editor. The reviewers have opted to remain anonymous.

The reviewers have discussed the reviews with one another and the Reviewing Editor has drafted this decision to help you prepare a revised submission.

Summary:

Gluth and colleagues present data from four experiments with an initial motivation to replicate preference reversals in economic choice in response to distractor values as described by Chau et al. (2014). None of these experiments provided evidence for these sorts of reversals. Careful statistical analysis and comprehensive quantitative modeling of the behavioral data combined with eye tracking provided an explanation for the discrepancies and suggest that distractor effects are best described in terms of value-based attentional capture. They further noted that some of the key effects reported by Chau et al. may have arisen as a statistical artifact. This paper is a very nice example of progress and self-correction in the scientific literature. Overall the experiments and analysis appear quite thorough.

Essential revisions:

The below points reflect a prioritized amalgamation of individual reviewer comments.

1) The experimental design and analyses rest heavily on the assumption that individual participants appropriately compute expected values from probability and value information, yet unless I missed it, there was no direct test of this notion. It would be useful to verify this, and also to show that the basic predictions and results hold for subjective value estimates as well as expected values.

2) The authors interpret the fixation duration as a measure of attention, but their data and that of some of the most relevant cited literature (Cavanagh et al., 2014) would be just as consistent with it providing a measure of intention to select a particular option. Under such an interpretation, its less clear whether the eye-tracking data provide selective evidence for the proposed attentional mechanisms.

3) The authors state several times that divisive normalization predicts a change in the relative choice accuracy/frequency between HV and LV options. The predictions for relative and absolute choice accuracy in this task are still a bit fuzzy for several models, but especially divisive normalization. Wouldn't divisive normalization also predict a decrease in absolute choice accuracy as the value of D increases? It would help to show both the relative and absolute accuracy predictions for all alternative models at least once (perhaps in Figure 1D).

I think that the authors should refer to the divisive normalization model in Louie et al. (2013) as "divisive normalization of integrated values" or some similar, but better term. The idea would be to make sure readers understand that normalization at the level of integrated attribute values is one specific form of normalization. Just as with the extension of the MIVAC, assuming attribute-level or hierarchical (divisive) normalization leads to key differences in model predictions.

Further, normalization is inherent in mutual inhibition models and I think that fact should be stated more clearly in the paper. It is important to separate the qualitative and quantitative distinctions between the specific form of divisive normalization proposed in Louie et al. (2013), from other forms of normalization (e.g. the weighted subtraction in competing accumulator models). However, it is also important not to equate divisive normalization, in general, with a specific instantiation from one paper.

4) Methods questions:

– In the logistic regression examining additive and interactive effects of value and fixation duration, the authors specify interactions between fixation and HV or LV separately, but include only a main effect for the value difference, HV-LV. I don't know if this coding of the two main effects as a contrast vector makes any difference in fitting or interpreting the model, but I think it would be worthwhile and easy to check.

– How were correlations between expected value and relative fixation duration aggregated across participants to generate the t and p-values listed in the Results? I did not find any details on this analysis in the Materials and methods section and so assume it was a standard parametric t-test. Is this parametric test appropriate for the distribution of *r*-values across participants?

---

## [Author Response]

Essential revisions:The below points reflect a prioritized amalgamation of individual reviewer comments.1) The experimental design and analyses rest heavily on the assumption that individual participants appropriately compute expected values from probability and value information, yet unless I missed it, there was no direct test of this notion. It would be useful to verify this, and also to show that the basic predictions and results hold for subjective value estimates as well as expected values.

We agree that our analyses rely on the approximation that people take the expected value into account when making their decisions. Therefore, we followed the reviewer’s suggestion and examined whether (and how) people integrate probability and outcome magnitude when making decisions by comparing the following five choice models against each other (with regard to how accurately they predict choices of HV vs. choices of LV; details can be found in the Materials and methods subsection “Testing the assumption of integration of magnitude and probability information”):

1) “Only Magnitude” (OM): A model that assumes choices are solely based on magnitude differences.

2) “Only Probability” (OP): A model that assumes choices are solely based on probability differences.

3) “Expected Value” (EV): A model that assumes choices are based on EV differences.

4) “Expected Utility” (EU): An extension of the EV model that assumes a power utility function (u[x] = x^α^; EU[x] = u[x] * p), with one additional free parameter allowing to represent risk preferences.

5) “Prospect Theory” (PT): An extension of the EU model that assumes that probabilities are weighted according to Cumulative Prospect Theory (Tversky and Kahneman, 1992), with one additional free parameter.

Comparing these models against each other (Figure 5—figure supplement 1A, left and middle panels) showed that the EV model provided a better account of the data than the single-attribute models (OM and OP). Only 15 of 123 participants were best explained by one of the two single-attribute models. Therefore, we can conclude that (the great majority of) participants integrated magnitude and probability when making decisions.

However, the comparison also revealed that the EU model explains choices better than the EV model (whereas the increased complexity of the PT compared to the EU model does not seem to be justified by the data). The distribution of the power utility parameter α of the EU model (Figure 5—figure supplement 1A, right panel) shows that for most participants the parameter lies in the range between 0 and 2, with some more extreme values to the right.

For reasons of simplicity and comparability to Chau et al. (2014), we kept using the EV-based value estimates for initial presentation of statistical and modeling analyses in the revised manuscript. However, given that the EU model provided a better account of the data than the EV model, we followed the reviewer’s suggestion and checked whether the central results would hold when replacing EVs by EU-based subjective value estimates. As can be seen in Figure 5—figure supplement 1, the central behavioral results of our study are largely unaffected by this change of analysis: The HV-D effect on relative choice accuracy remains insignificant; the HV-D effect on absolute choice accuracy remains significantly positive, as do the effect of D’s value on choices of D and all RT-related effects. Only the effect of D’s value on too-slow errors does not reach significance anymore (*p* =.077).

We also tested whether the predictions of MIVAC would change. Thereto, we reran the model using the EU-based subjective value estimates instead of the EVs as inputs to the model (for details, see the Materials and methods subsection “Modeling procedures for MIVAC IV: MIVAC with subjective values as inputs”). As can be seen in Figure 7—figure supplement 2, the predictions of the model do not change qualitatively.

Also note that the (qualitative) predictions of the different models with respect to the novel trials remain the same. That is, as long as we can be certain that people did not solely rely on only one of the attributes (which the above-mentioned model comparison clearly suggests), the predictions regarding the multi-attribute space (see Figure 1C and 1D) as well as the analyses (Figure 5D) remain unchanged.

2) The authors interpret the fixation duration as a measure of attention, but their data and that of some of the most relevant cited literature (Cavanagh et al., 2014) would be just as consistent with it providing a measure of intention to select a particular option. Under such an interpretation, its less clear whether the eye-tracking data provide selective evidence for the proposed attentional mechanisms.

We agree with the reviewer that it can be difficult to disentangle whether people look at choice options to pay attention to them (to accumulate evidence about the option’s value) or whether they look at it because they intend to choose it. In the revised manuscript, we discuss these two possibilities (Discussion, fourth paragraph).

Compared to previous studies (Cavanagh et al., 2014; Shimojo et al., 2003), however, we believe that our results provide convincing evidence for interpreting fixation duration as a measure of attention and – more importantly – as driving a biased evidence accumulation process (i.e., the mechanism implemented in MIVAC).

First of all, we reported in the original manuscript that the first fixation was influenced by value. This is consistent with value-based attentional capture but difficult to explain by assuming that participants developed an intention to choose an option already before the first fixation. In the revised manuscript, we show that the influence of value on first fixations remains significant (for options HV and D) even after controlling for the eventual choice (Materials and methods, subsection “Eye-tracking analysis III: analysis of initial fixations”, first paragraph).

Second, we added an analysis testing whether the first fixation and its duration predicted choice over and above the value difference between HV and LV and the value of D. These analyses yielded significant effects of the first fixation and its duration as well as of value information (Materials and methods, subsection “Eye-tracking analysis III: analysis of initial fixations”, last paragraph). Stated differently, whether and how long an option is fixated first predicts choice, but it is not a perfect and not the only predictor. Rather, fixating an option (gradually) increases the probability of choosing an option, consistent with its implementation as a bias-in-evidence-accumulation mechanism in MIVAC.

3) The authors state several times that divisive normalization predicts a change in the relative choice accuracy/frequency between HV and LV options. The predictions for relative and absolute choice accuracy in this task are still a bit fuzzy for several models, but especially divisive normalization. Wouldn't divisive normalization also predict a decrease in absolute choice accuracy as the value of D increases? It would help to show both the relative and absolute accuracy predictions for all alternative models at least once (perhaps in Figure 1D).

In the revised manuscript, we show the predictions of the alternative models with regard to absolute choice accuracy in Figure 1—figure supplement 1. It is correct that the divisive normalization model also predicts a decrease in absolute choice accuracy as the value of D increases (as long as the model treats the distractor as a potential choice option). In fact, all models make this prediction (although for the biophysical cortical attractor network model proposed by Chau et al., 2014, it remains unclear whether the positive effect on relative choice accuracy is reduced, evened out, or reversed by the negative effect on absolute choice accuracy).

Therefore, it is not the presence of an effect on absolute choice accuracy but the absence of an effect on relative choice accuracy in our data that speaks against the divisive normalization model.

I think that the authors should refer to the divisive normalization model in Louie et al. (2013) as "divisive normalization of integrated values" or some similar, but better term. The idea would be to make sure readers understand that normalization at the level of integrated attribute values is one specific form of normalization. Just as with the extension of the MIVAC, assuming attribute-level or hierarchical (divisive) normalization leads to key differences in model predictions.Further, normalization is inherent in mutual inhibition models and I think that fact should be stated more clearly in the paper. It is important to separate the qualitative and quantitative distinctions between the specific form of divisive normalization proposed in Louie et al. (2013), from other forms of normalization (e.g. the weighted subtraction in competing accumulator models). However, it is also important not to equate divisive normalization, in general, with a specific instantiation from one paper.

In the Introduction of the revised manuscript, we make clear that we refer exclusively to the model by Louie et al. (2013) and give references to work showing that hierarchical normalization models can make qualitatively different predictions (Hunt et al., 2014; Landry and Webb, 2017; Soltani et al., 2012). We also mention that divisive normalization is a much more general (“canonical”) neural computation, and that the model proposed by Louie et al. (2013) is only one specific instantiation of this principle (Introduction, third paragraph).

Furthermore, we discuss the relationship between normalization and mutual inhibition (Discussion, sixth paragraph). In our view, it would not be fully correct to equate normalization with the principle of mutual inhibition/weighted subtraction of competing accumulators. Note that Teodorescu et al. (2016) compared normalization-based and LCA-based models against each other and showed that these models can make different predictions in particular with respect to response times. According to Teodorescu et al., normalization occurs at the level of input while mutual inhibition occurs at the accumulator level.

4) Methods questions:– In the logistic regression examining additive and interactive effects of value and fixation duration, the authors specify interactions between fixation and HV or LV separately, but include only a main effect for the value difference, HV-LV. I don't know if this coding of the two main effects as a contrast vector makes any difference in fitting or interpreting the model, but I think it would be worthwhile and easy to check.

We repeated the regression analysis using two separate predictor variables for HV and LV instead of a single HV-LV predictor variable. Expectedly, HV is positively linked to accuracy and LV is negatively linked to accuracy in this analysis. More importantly, the effects of the remaining predictor variables are unaffected by this change of analysis. In the revised manuscript, we replaced the old analysis by this new analysis in Figure 4—figure supplement 1 but mention that using a single HV-LV predictor variable yields equivalent results.

– How were correlations between expected value and relative fixation duration aggregated across participants to generate the t and p-values listed in the Results? I did not find any details on this analysis in the Materials and methods section and so assume it was a standard parametric t-test. Is this parametric test appropriate for the distribution of r-values across participants?

The reviewer is correct that one-sample *t*-tests were used to test whether the correlation coefficients between value and fixations differed from 0. Using the Kolmogorov-Smirnoff tests, we can confirm that the distributions of these coefficients do not differ significantly from a normal distribution (at *p* <.1). However, because correlation coefficients are bounded between -1 and 1, we adapted our analysis for the revised manuscript: Correlation coefficients were first transformed via Fisher z-transformation, and these transformed values were then subjected to Kolmogorov-Smirnoff tests (again not significant at *p* <.1) and one-sample *t*-tests against 0. Thus, the values shown in Figure 4A are still correlation coefficients (i.e., *r*-values) but the statistical analysis are based on Fisher z-transformed values. These changes are mentioned in the Materials and methods subsection “Eye-tracking analysis I: tests of value-based attentional and oculomotor capture”, and the Results are updated accordingly (subsection “Experiment 3: value affects attention and attention affects decisions”, last paragraph).